# Recent glacier and lake changes in High Mountain Asia and their relation to precipitation changes

Désirée Treichler[1], Andreas Kääb[1], Nadine Salzmann[2], and Chong-Yu Xu[1]

[1]Department of Geosciences, University of Oslo, Sem Sælands vei 1, 0371 Oslo, Norway
[2]Department of Geosciences, University of Fribourg, Chemin du Musée 4, 1700 Fribourg, Switzerland

**Correspondence:** Désirée Treichler (desiree.treichler@geo.uio.no)

**Abstract.** We present an updated, spatially resolved estimate of 2003–2008 glacier surface elevation changes for entire High Mountain Asia (HMA) from ICESat laser altimetry data. The results reveal a diverse pattern that is caused by spatially greatly varying glacier sensitivity, in particular to precipitation availability and changes. We introduce a spatially resolved zonation where ICESat samples are grouped into units of similar glacier behaviour, glacier type, and topographic settings. In several regions, our new zonation reveals local differences and anomalies that have not been described previously. Glaciers in the Eastern Pamirs, Kunlun Shan and central TP were thickening by 0.1–0.7 m a$^{-1}$, and the thickening anomaly has a crisp boundary in the Eastern Pamir that continues just north of the central Karakoram. Glaciers in the south and east of the TP were thinning, with increasing rates towards southeast. We attribute the glacier thickening signal to a stepwise increase in precipitation around ∼1997–2000 on the Tibetan Plateau (TP). The precipitation change is reflected by growth of endorheic lakes in particular in the northern and eastern TP. We estimate lake volume changes through a combination of repeat lake extents from Landsat data and shoreline elevations from ICESat and the SRTM DEM for over 1300 lakes. The rise in water volume contained in the lakes corresponds to 4–25 mm a$^{-1}$, when distributed over entire catchments, for the areas where we see glacier thickening. The precipitation increase is also visible in sparse in-situ measurements and MERRA-2 climate reanalysis data, but less well in ERA Interim reanalysis data. Taking into account evaporation loss, the difference between average annual precipitation during the 1990s and 2000s suggested by these datasets is 34–100 mm a$^{-1}$, depending on region, which can fully explain both lake growth, and glacier thickening (Kunlun Shan) or glacier geometry changes such as thinning tongues while upper glacier areas were thickening or stable (eastern TP). The precipitation increase reflected in these glacier changes possibly extended to the northern slopes of the Tarim Basin, where glaciers were nearly in balance in 2003–2008. Along the entire Himalaya, glaciers on the first orographic ridge, which are exposed to abundant precipitation, were thinning less than glaciers in the dryer climate of the inner ranges. Thinning rates in the Tien Shan vary spatially but are rather stronger than in other parts of HMA.

## 1 Introduction

High Mountain Asia (HMA) is a large and remote region hosting a range of topographic and climatic regimes (Palazzi et al., 2013). Some areas, like the Himalaya or Karakoram, are characterized by steep orographic gradients (Bolch et al., 2012).

Glacier landscape and shapes, climate, elevation, and consequently glacier behaviour and response to climate change, vary strongly throughout the region (e.g. Scherler et al., 2011; Fujita and Nuimura, 2011; Bolch et al., 2012; Brun et al., 2017; Sakai and Fujita, 2017). Throughout the recent decades, most glaciers in the region seem to have lost mass and retreated (e.g. Bolch et al., 2012; Kääb et al., 2012; Brun et al., 2017). Yet, there are some exceptions, most prominent the so-called Karakoram or Pamir-Karakoram anomaly (e.g. Hewitt, 2005; Quincey et al., 2011; Kääb et al., 2012; Gardelle et al., 2013; Kapnick et al., 2014), and positive mass balances are reported for some glaciers on the Tibetan Plateau (TP) and Kunlun Shan (Yao et al., 2012; Kääb et al., 2015; Brun et al., 2017).

At the same time, a number of studies report expansion of endorheic lakes on the TP starting around the beginning of this century (e.g. Zhang et al., 2017). For these lake systems, additional lake water masses either stem from increased lake inflow, i.e. mainly increased precipitation or enhanced glacier melt, or from reduced water loss, i.e. mainly decreased evaporation. However, in-situ meteorological data, which could shed light on precipitation and evaporation changes and their spatial patterns, are barely available for the HMA (Kang et al., 2010) and lacking in particular for the remote areas on the TP and Kunlun Shan with suggested recent positive glacier mass balances. In addition, in-situ measurements at high altitude, in particular for precipitation, are in general subject to challenges (Salzmann et al., 2014). These scarceness and problems associated with in-situ measurements likely also affect the accuracy and reliability of reanalysis data over some zones of HMA, leaving thus an overall limited understanding of glacier changes and associated climate changes over significant areas of HMA.

HMA region-wide assessments of glacier changes have been derived either from (i) interpolating the sparse in-situ measurements (Cogley, 2011; Bolch et al., 2012; Yao et al., 2012), (ii) digital elevation model (DEM) differencing (Gardelle et al., 2013; Brun et al., 2017), (iii) GRACE (Gravity Recovery and Climate Experiment) gravimetry data (Matsuo and Heki, 2010; Jacob et al., 2012; Gardner et al., 2013), or (iv) ICESat satellite laser altimetry (Kääb et al., 2012; Gardner et al., 2013; Neckel et al., 2014; Kääb et al., 2015; Phan et al., 2017; Brun et al., 2017). Of these, only Brun et al. (2017), Gardner et al. (2013) and the coarse-resolution GRACE studies cover the entire HMA, including also Tien Shan, TP and Qilian Shan. For some regions, the differences between the studies are considerable, even if they address the same time period (Cogley, 2012; Kääb et al., 2015). All four method principles listed above have their specific advantages and disadvantages. A challenge with GRACE data, for instance, is the separation of mass changes due to glacier mass loss and other influences, such as changes in lake and ground water storage (e.g. Baumann, 2012; Yi and Sun, 2014). For some DEM differencing studies in the region, a major source of uncertainties is the Shuttle Radar Topography Mission (SRTM) DEM. The SRTM DEM is based on C-band radar that can penetrate up to several metres into snow and ice, depending on the local snow and ice conditions during the SRTM data acquisition in February 2000 (Gardelle et al., 2012b; Kääb et al., 2015). The recent study of Brun et al. (2017) is not affected by radar penetration as it is exclusively based on time series from ASTER optical stereo DEMs. While their new data set of time-averaged geodetic glacier mass balances is spatially of unprecedented extent and detail, ASTER DEMs suffer from limitations such as sensor shaking (jitter) (Girod et al., 2017), biased errors/voids in particular in featureless accumulation areas (Wang and Kääb, 2015; McNabb et al., 2019), and spatio-temporal variations in image acquisitions (Berthier et al., 2016; Brun et al., 2017) that cause the studied time periods to vary throughout the area. The study of Brun et al. (2017) includes a comparison to ICESat 2003–2008 surface elevation changes, although using large spatial regions and ASTER DEMs from

2000–2008 as ASTER DEM stacks were too noisy for shorter time spans. With in-situ measurements and ICESat laser data, the uncertainty lies in the representativeness of the spatial sampling. Both are not spatially continuous but sample only some glaciers, although ICESat with higher density of footprints than in-situ measurements. Direct mass balance measurements are only available for few glaciers (WGMS, 2016, Fig. 1), and the overall mass balance signal they suggest is possibly biased towards glaciers at low elevations because these are easier to access (Wagnon et al., 2013).

From recent glacier studies involving ICESat data over HMA, Kääb et al. (2015) suggest that results are sensitive to zone delineation, in particular in areas with strong spatial variability of glacier thickness changes. Studies stress the importance of sampling the glacier hypsometry correctly, i.e. that the number of data points per elevation reflects the glacierised area at each elevation. Kääb et al. (2012, 2015) and Treichler and Kääb (2016) found that hypsometries of individual years of ICESat samples may not fit the glacier hypsometry, even if the total sample base from the entire studied period reflects the glacier elevation distribution accurately. This can alter the results in cases where there is a consistent elevation trend in sampling elevations, i.e. the average sampling elevation increases or decreases over the studied time period. Correct and up-to-date glacier outlines turn out to be very important for deriving ICESat elevation changes. Inclusion of non-glacier elevation measurements, where surface elevation is stable, reduces the glacier elevation change retrieved from ICESat. The effect of snow cover, and thus the choice of whether including ICESat winter campaigns or not, plays a role — also for the autumn 2008 ICESat campaign that was completed in December 2008 only due to technical problems (Kääb et al., 2012; Gardner et al., 2013; Treichler and Kääb, 2016). Spatially varying vertical biases from DEMs used as reference can considerably increase trend uncertainty (Treichler and Kääb, 2016). All ICESat studies in HMA so far rely on the SRTM DEM, where spatially varying penetration could be a source of such biases.

The present study has two objectives. First, we aim to extend the ICESat-based work of Kääb et al. (2012, 2015) to entire HMA, including the Tibetan Plateau, Qilian Shan and Tien Shan, and under special consideration of the issues addressed above and the recent method improvements by Treichler and Kääb (2016). In particular, we present a new elevation change zonation into spatial units that consider glacier topo-climatic setting, behaviour and type rather than relying on a regular grid or Randolph Glacier Inventory (RGI Consortium, 2017) regions. Second, we investigate the possible cause of the positive glacier volume changes in the TP and Kunlun Shan regions with the hypothesis of a precipitation increase in this area. For the latter purpose, we quantify the water volume changes in endorheic lakes on the TP, their timing and spatial pattern, and set them in relation to the independent ICESat-derived glacier surface elevation changes as well as precipitation estimates from climate reanalyses and sparse in-situ measurements from meteorological stations.

## 2 Study region

The HMA glacier region is covered by about 100'000 km$^2$ of glacier area (RGI Consortium, 2017). Temperature rise due to global climate change is especially pronounced on the TP and increasing with elevation (Liu and Chen, 2000; Qin et al., 2009; Ran et al., 2018). Glaciers are found on all large mountain ranges around the TP at $> 4000$ m a.s.l. but mostly to the south and west, where the steep elevation gradient from the Indian planes acts as a barrier for moisture that is advected by

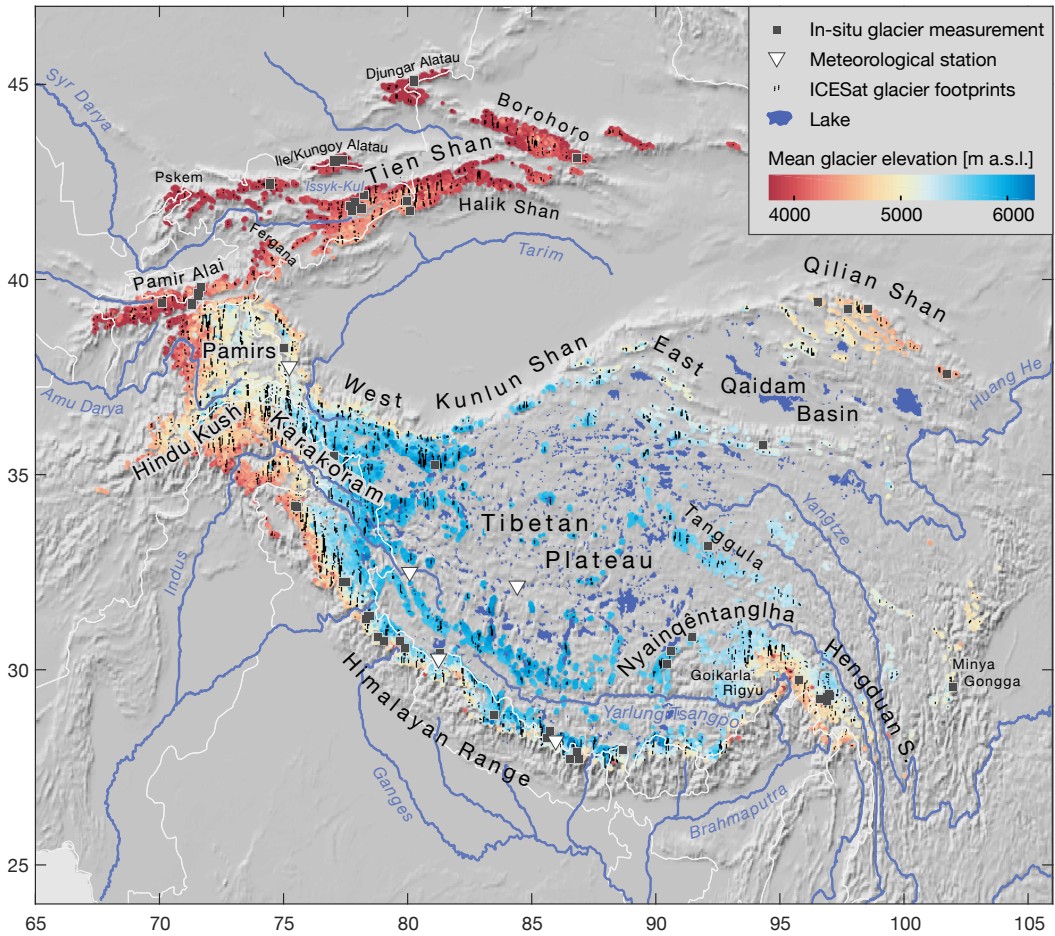

**Figure 1.** Mountain ranges and major rivers in High Mountain Asia, with meteorological stations (triangles), lakes on the TP and in the Qaidam Basin (dark blue) and ICESat glacier samples (black dots) used in this study. Dark grey squares show all in-situ glacier mass balance measurements done at some point during the last decades (WGMS, 2016), and RGI glaciers are coloured according to their mean elevation.

the Indian monsoon (Himalaya, Karakoram, Eastern Nyainqêntanglha Shan) and Westerlies (Hindu Kush, Karakoram, Pamir), respectively (Yao et al., 2012; Bolch et al., 2012; Mukhopadhyay and Khan, 2014). On the very dry TP, glaciers occur only on the sparsely spread small mountain ranges.

In interplay with the Siberian High further north (Narama et al., 2010; Böhner, 2006), the Westerlies are the dominant
5 source of moisture for the mountains surrounding the dry Tarim basin (at ca. 1000 m a.s.l.) — the Tien Shan to the north, and Kunlun Shan to the south (Ke et al., 2015; Yao et al., 2012). The mountain ranges at the eastern margins of the TP (Qilian Shan, Hengduan Shan, Minya Gongga) are also influenced by the East Asian Monsoon (Yao et al., 2012; Li et al., 2015). In both the monsoonal and westerly regimes, precipitation decreases northward (Bolch et al., 2012). Depending on the regionally dominant source of moisture, glacier accumulation happens at different times of the year (Bolch et al., 2012; Maussion et al.,

2014; Yao et al., 2012; Sakai et al., 2015). From the eastern Himalaya and southern/eastern TP to the northwest of HMA, there is a transition from predominant spring/summer accumulation to winter accumulation in the Hindu Kush and the western parts of Tien Shan (Palazzi et al., 2013; Bookhagen and Burbank, 2010; Rasmussen, 2013). Mountains in between, such as the Karakoram and western Himalaya, receive moisture from both sources (Kuhle, 1990; Bolch et al., 2012). The Kunlun
Mountains, on the other hand, receive most precipitation around May (Maussion et al., 2014).

For the HMA glaciers with predominant spring/summer accumulation, glacier accumulation and ablation happen at the same time. Besides rising temperatures, recent studies suggest that climate change and altered circulation patterns affect radiation regimes and thus also glacier ablation in HMA through, e.g., changes in evapotranspiration or cloud cover (Forsythe et al., 2017; de Kok et al., 2018). The seasonal timing of snow accumulation on glaciers thus likely plays an important role for
glacier sensitivity to a warming climate (Fujita, 2008; Mölg et al., 2012; Sakai and Fujita, 2017). Another important factor is total precipitation which depends on continentality (Shi and Liu, 2000; Kuhle, 1990) and, on smaller spatial scale, on glacier location on or behind a mountain range that acts as a primary orographic barrier or causes orographic convection. Wagnon et al. (2013) and Sherpa et al. (2016) found indications of steep horizontal precipitation gradients within only a few kilometres on the outermost ridge of the Great Himalaya in the Khumbu region in Nepal. Vertical precipitation gradients at high altitude
are still poorly understood. It is suggested that precipitation increases from dry mountain valley bottoms to an elevation of 4000–6000 m a.s.l. and subsequently decreases again at even higher elevations (e.g. Immerzeel et al., 2014, 2015).

Many glaciers in HMA are debris-covered in their ablation areas, and the percentage of debris-covered ice varies greatly between different regions (Scherler et al. 2011; Gardelle et al. 2013, Kraaijenbrink et al. 2017, suppl.). Recent studies have found that although debris-covered glaciers in HMA have stable front positions (Scherler et al., 2011), they melt on average
just as fast as clean ice glaciers (Kääb et al., 2012; Gardelle et al., 2012b; Pellicciotti et al., 2015). Thus, in this study, we do not explicitly distinguish between debris-covered and debris-free glacier tongues.

## 3   Data and Methods

In this section we give a short overview of the data and methods used. Details can be found in the Appendix.

### 3.1   Data

For deriving repeat elevations on glaciers and lakes, we use data from the NASA Geoscience Laser Altimeter System (GLAS) aboard the Ice, Cloud and land Elevation Satellite (ICESat) that measured the Earth's surface elevations in two to three campaigns per year from 2003 to 2009 (Zwally et al., 2012, GLAH14). The campaigns were flown in northern autumn (∼October–November), winter (∼March), and early summer (∼June). (Appendix A1).

As reference DEM for our ICESat processing and to derive lake shoreline elevations we use the DEM from the Shuttle
Radar Topography Mission (SRTM, Farr et al., 2007; Farr and Kobrick, 2000). We used the C-band, non-void-filled SRTM DEM version at 3 arc-seconds resolution (SRTM3). As an alternative elevation reference, we used also the SRTM DEM at 1-arc-second resolution (SRTM1). (Appendix A2).

As an estimate for regional and temporal precipitation patterns for the years 1980–2015 we use data from the reanalysis products MERRA-2 (Gelaro et al., 2017) and ERA Interim (Dee et al., 2011). We use monthly summarised values of the variables total precipitation, snowfall and evaporation. The two chosen reanalysis products have previously been found to model precipitation comparatively well in our study area (Chen et al., 2019; Cuo and Zhang, 2017; Sun et al., 2018). (Appendix A3). We did not use the recently released, higher-resolution ERA5 data since the data assimilation scheme behind that product includes less forcing data relevant for modelling precipitation at high elevations (above 1500 m asl) — i.e. entire HMA — compared to the older ERA Interim (Orsolini et al., 2019).

Further, we use in-situ data from the five westernmost meteorological stations on the TP and Kunlun Shan (Fig. 1), provided by the China Meteorological Science Data Sharing Service Network. The data include daily measurements of precipitation, mean air temperature, and for the four stations on the southwestern TP also evaporation.

We extract repeat lake coverage from the Global Surface Water dataset (Pekel et al., 2016) that is a classification of the entire Landsat archive into monthly and annual maps of surface water. The data are available within Google Earth Engine. Spatial coverage is nearly complete (>98%) starting from 2000 but considerably worse for some years of the 1990s. (Appendix A4).

### 3.2 Methods for glacier volume change

We use surface elevation measurements from ICESat data points on glaciers and surrounding stable terrain and follow the double-differencing method explained in further detail in Kääb et al. (2012) and Treichler and Kääb (2016), with special consideration of issues mentioned in the above introduction (Appendix B). The difference between ICESat and SRTM elevations is further referred to as dh. Double differencing, i.e. fitting a linear trend through dh from several years, reveals how much the surface elevation has changed on average over the time period studied. We used only samples from ICESat's 2003–2008 autumn campaigns, the season with least snow cover in entire HMA, to avoid bias from temporal variations in snow depths (see introduction). After filtering, 74'938 glacier samples and about ten times as many off-glacier samples remain. ICESat data need to be grouped into spatial units to receive surface elevation changes. The samples within each spatial unit need to reflect the glaciers in a representative way — which means that the spatial units need to be chosen such that they group glaciers that are similar to each other in terms of climatic and topographical attributes, including their 2003–2008 mass balances and variations thereof. Previous studies have used regular grids, the RGI regions, or their own arbitrary zonation. These do not necessarily fulfil the above requirements. We considered automated clustering methods to receive spatial units from ICESat dh directly but were not successful. We therefore preferred to delineate spatial units manually, considering topographic and climatic setting, elevation, visual glacier appearance, and input from literature and discussions with experts (Appendix B1). In particular, we paid attention to orographic barriers. The zonation we present here is thus the result of an iterative manual process of re-defining spatial units until they satisfied these criteria. After computing linear regressions on glacier dh, we split or merged some of the previously drawn units such that the final zonation yielded statistically stable and robust glacier surface change estimates. While the procedure is based on carefully applied expert knowledge, we are fully aware that our zonation is eventually a subjective one and certainly open to discussion. As a control approach, we applied the same gridding method as Kääb et al. (2012, 2015) to the entire HMA.

It is very important to ensure ICESat's elevation sampling is consistent through time and representative for glacier hypsometry (see introduction). We apply four different ways of correcting hypsometry mismatches of ICESat sampling (Appendix B2). Per spatial unit, we estimate glacier surface elevation change by fitting a robust linear regression through individual dh (which minimises an iteratively weighted sum of squares) and also compute a t-fit (Treichler and Kääb, 2016) and a non-parametric

Theil-Sen linear regression (Theil, 1950; Sen, 1968). Our 'standard method' for the final glacier elevation change estimates corresponds to the average of all hypsometry-correcting methods and linear regression methods. Additionally, we also compute elevation change for only the upper/lower 50% glacier elevations as from RGI hypsometries (samples above/below the median RGI glacier elevation of each individual glacier) for each spatial unit. The latter analysis violates mass conservation and should thus not be interpreted in terms of mass balance, but rather, for instance, for changes in glacier elevation gradients

(e.g. Brun et al., 2017; Kääb et al., 2018). To allow comparison with other glacier studies and changes in lake and precipitation water masses, we use RGI glacier areas to convert our surface elevation change rates to volume/mass changes (Supplementary Information S2).

Glacier dh may be subject of vertical bias from elevation differences that are caused by other reasons than glacier surface elevation change, i.e. from bias in the local reference elevation (the SRTM DEM) or snow fall. We compute corrections for

these biases (Appendix B3). Local vertical bias may result from inconsistent reference DEM age or production, tiling and tile/scene misregistration, or locally varying radar penetration (in case of the SRTM DEM). To remove this bias, we compute a per-glacier elevation correction cG, corresponding to the median dh for each glacier, according to the method described in Treichler and Kääb (2016). Treichler and Kääb (2016, 2017) found that ICESat clearly records the onset of winter snowfall in Norway during the split autumn 2008 campaign (stopped half way in mid-October and completed only in December).

Analogue to Treichler and Kääb (2016), we estimate December 2008 snow bias from a linear regression of October/December 2008 off-glacier dh on elevation and time.

### 3.3    Methods for lake volume change

We derive the volume changes of lakes on the endorheic TP in order to relate glacier changes and precipitation changes on the Tibetan Plateau to each other. In particular, we want to investigate whether precipitation increases could be a reason for

the positive glacier mass balances found in parts of the region. For endorheic lake systems, additional lake water masses either stem from increased lake inflow (mainly increased precipitation and enhanced glacier melt, possibly also thawing permafrost and changes in groundwater storage) or from reduced water loss (mainly changes in evaporation). This section provides a summary of the methods, details can be found in Appendix C.

We compute annual water volume change of the Tibetan lakes by multiplying annual lake areas with water level changes

from repeat water surface elevations for each year over the period 1990–2015. Maximum annual lake extents are obtained directly from the Global Surface Water data set. We retrieve the corresponding lake surface elevations in two ways: a) from SRTM DEM elevations of the lake shore by computing the median of interpolated DEM elevations for lake shore cells for each areal extent, and b) directly from ICESat footprint elevations on the lake areas for those lakes where ICESat data are available. The two datasets used have different strengths: ICESat-derived lake surface elevations are far more accurate but available only

for about a tenth of all lakes. To extend the lake elevation time series from method b) beyond the ICESat period of 2003–2009, we compute the area–surface-elevation relationship for each lake by robust linear regression and apply this function to the areal extends of the years before and after the ICESat period, both for annual timeseries and individual ICESat campaigns. The so-extrapolated surface elevation values generate complete 1990–2015 time series for both areal extent and lake levels from

SRTM and ICESat data, respectively. Our method is in parts similar to the methods used by previous studies (e.g. Zhang et al., 2017) but the inclusion of a DEM for deriving shoreline elevations, and thus lake water levels in addition to altimetry data, enabled us to produce volume change time series for one order of magnitude more lakes (>1300) than derived previously.

To minimise the effect of uncertainties in or erroneous estimates for individual years, we analyse time series in a summarised way through regression over time and as decadal averages, and apply a range of filters. (Appendix C1). To estimate the lake

water volume change in a way that can be related to glacier mass balances and precipitation changes (i.e. mm w.e. per m$^2$), we summarise and spatially distribute the water volume changes of all lakes within spatially confined basins aggregated from the endorheic catchments of the USGS HydroSHEDS dataset (Lehner and Döll, 2004). (Appendix C2).

## 4   Results

### 4.1   Glacier thinning and thickening

Figure 2a shows the 100 spatial units of glacier surface elevation change that result from the iterative manual zone delineation process. Spatial units needed to be large on the TP where glacier density is low, and could be rather small in the Karakoram which is intensely glacierised. Along major ridges such as the Himalaya, the units were designed narrow and along ridge orientation in order to group glaciers under similar temperature and precipitation regimes rather than across orographic barriers.

Surface elevation change for the new spatial units and the $2° \times 2°$ grid in Fig. 2b are derived using the 'standard method'

except for 34 units with hypsometry missampling or elevation bias (Appendix B). The error values given in Fig. 2c and in the text conservatively include, where applicable, uncertainties from off-glacier elevation trends, the deviation from the standard method (greatly increased errors, units showing up in yellow in Fig. 2c), and December 2008 snow fall correction (Supplementary Information S1). In areas with snow-rich winters, the latter may contribute up to 40% of the error budget. In Fig. 2b, the size of the circles corresponds to the number of samples (minimum 200) while the overlaid, grey circles show

the trend error (at $1\sigma$) in relation to the trend slope; i.e. elevation changes are not statistically significantly different from zero where the grey circles fully cover the underlying coloured circle.

The overall pattern of elevation change is the same for both spatial zonation approaches; positive glacier elevation change in the Kunlun Shan an the inner TP, and spatially varying but modest glacier surface lowering in most areas except for very negative values in Nyainqêntanglha Shan/Hengduan Shan and parts of the Tien Shan. Most of HMA's glaciers seem to experi-

ence thinning both in their ablation and accumulation areas, as shown in Figs. 3a and 3b (upper and lower 50% of glacier area, respectively). Exceptions to these are the areas with positive glacier changes plus parts of the Himalayas and the mountains surrounding the Tarim basin, where upper glacier elevations seem relatively stable.

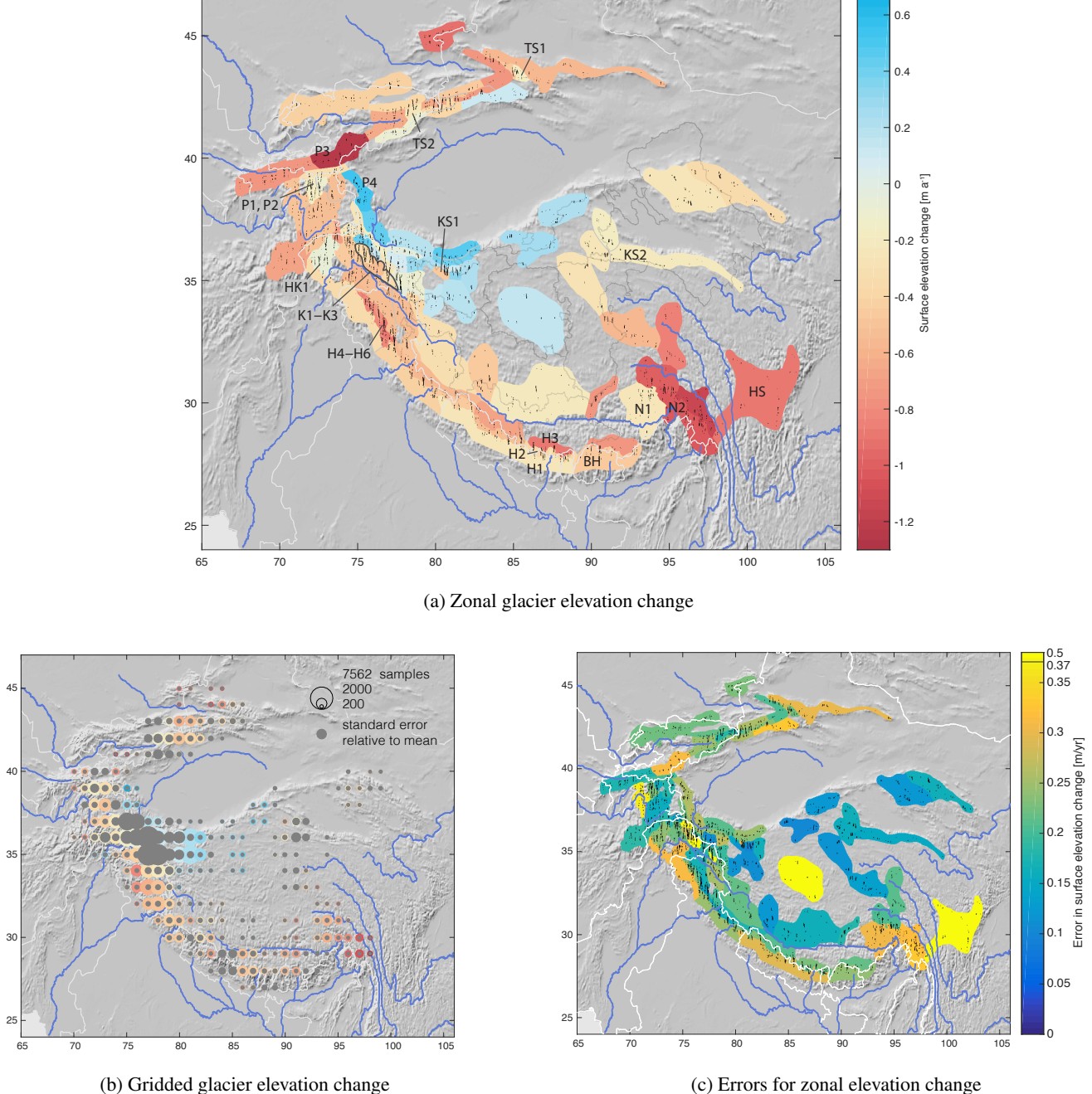

(a) Zonal glacier elevation change

(b) Gridded glacier elevation change

(c) Errors for zonal elevation change

**Figure 2.** 2003–2008 glacier elevation change rates for (a) manually delineated zones and (b) overlapping $2° \times 2°$ degree grid cells with $1°$ spacing. Colour bar (b) as in (a). Circles in (b) are scaled according to sample numbers. The overlaid grey circles show the standard error in relation to the slope of the linear fit, i.e. elevation change is not significantly different from zero (at $1\sigma$) where the coloured circles are fully covered. (c) Error for (a) at $1\sigma$, including uncertainties from deviations from the standard method, December 2008 snow fall correction, and trends in off-glacier samples. The four bright yellow units have uncertainties between 0.43–0.50 m a$^{-1}$.

While the grid zonation (and also the smaller grid cells of Brun et al. 2017) shows smooth transitions between areas of positive or negative glacier evolution, our zoned map suggests rather greater spatial variability and sharper boundaries of clusters of similar elevation change. The regular grid size is too small to reach minimum sample numbers in areas with sparse glacier coverage (TP, outer Hengduan Shan, parts of Tien Shan), and the signal from grid cells with few samples is spatially

less consistent than what the manually delineated, larger units suggest. The small units in the Karakoram and Kunlun Shan, on the other hand, reveal locally varying signals that are averaged out or not significant in the coarser grid zonation (e.g. units K1–K3 and KS1 in Fig. 2a). Our new zonation, surface elevation changes and corresponding glacier mass changes in Gt a$^{-1}$ (using RGI glacier areas, Supplementary Information S2) are available as a data supplement.

In the Himalaya, the manual zone delineation shows a clear transition from moderately negative elevation change on the first,

southern orographic ridge ($-0.15$ to $-0.34$ m a$^{-1}$, maximum trend error: $0.31$ m a$^{-1}$) compared to glaciers located further back to the north and on the edge of the TP ($-0.33 \pm 0.22$ m a$^{-1}$ to $-0.85 \pm 0.14$ m a$^{-1}$). This pattern (e.g. units H1, H2, H3) is consistent along the entire range except for the Bhutanese Himalaya, where ICESat's sampling pattern required grouping of several orographic ridges which together show stronger surface lowering (unit BH, $-0.40 \pm 0.24$ m a$^{-1}$). In the gridded zonation, the pattern becomes smoothed out and is thus not visible.

Glaciers in the inner Hindu Kush (HK1, $0.03 \pm 0.24$ m a$^{-1}$) and the highest regions of the Pamir (P1 $-0.07 \pm 0.23$, P2 $-0.03 \pm 0.16$ m a$^{-1}$) were close to balance during 2003–2008 while all surrounding units in the area show stronger glacier surface lowering. Similarly, the glaciers around Lhasa (Goikarla Rigyu, unit N1) lowered their surface by only $-0.18 \pm 0.31$ m a$^{-1}$ which is considerably less than the surrounding units and in particular the very negative values in East Nyainqêntanglha Shan/Hengduan Shan ($-0.96$ to $-1.14 \pm 0.33$ m a$^{-1}$).

Further towards the inner TP and in the Qilian Shan, surface lowering decreases to $-0.1$ to $-0.3 \pm 0.16$ m a$^{-1}$. In the central and northern parts of the TP and the Kunlun Shan it turns positive — for nearly all units $> 0.25$ m a$^{-1}$, to as much as $0.79 \pm 0.26$ m a$^{-1}$ in the Eastern Pamirs/Kongur Shan (P4). The boundary between positive and negative surface elevation change seems to be formed by the Muji Basin, upper Gez river and Tashkurgan Valley. All units to the north of the central Karakoram range were in balance or thickening. The glaciers of the central Karakoram range and southwest of it showed

moderate thinning ($-0.22$ to $-0.47 \pm 0.43$ m a$^{-1}$). In the Western Kunlun Shan region, surface elevation changes of the lower 50% elevations are more positive than those of the upper 50% elevations (Figs. 3a and 3b). This behaviour is visible for 13 units centred around KS1.

Interestingly, also glaciers on the northern edge of the Tarim basin seem to be closer to balance ($-0.3 \pm 0.26$ to $+0.21 \pm 0.33$ m a$^{-1}$) than those in more central or northern ridges of the Tien Shan. In the Tien Shan, most spatial units indicate glacier

surface lowering between $-0.35$ and $-0.8 \pm 0.25$ m a$^{-1}$, but two units with higher glacier elevations stick out due to their more moderate surface lowering; TS1: $-0.1 \pm 0.21$ m a$^{-1}$, and TS2: $-0.18 \pm 0.18$ m a$^{-1}$. Several other units right next to these have considerably more negative values. At the transition between Pamir and Tien Shan (P3), glacier surface elevation decreased by as much as $-1.23 \pm 0.31$ m a$^{-1}$ — despite the thickening signal just south and east of this unit.

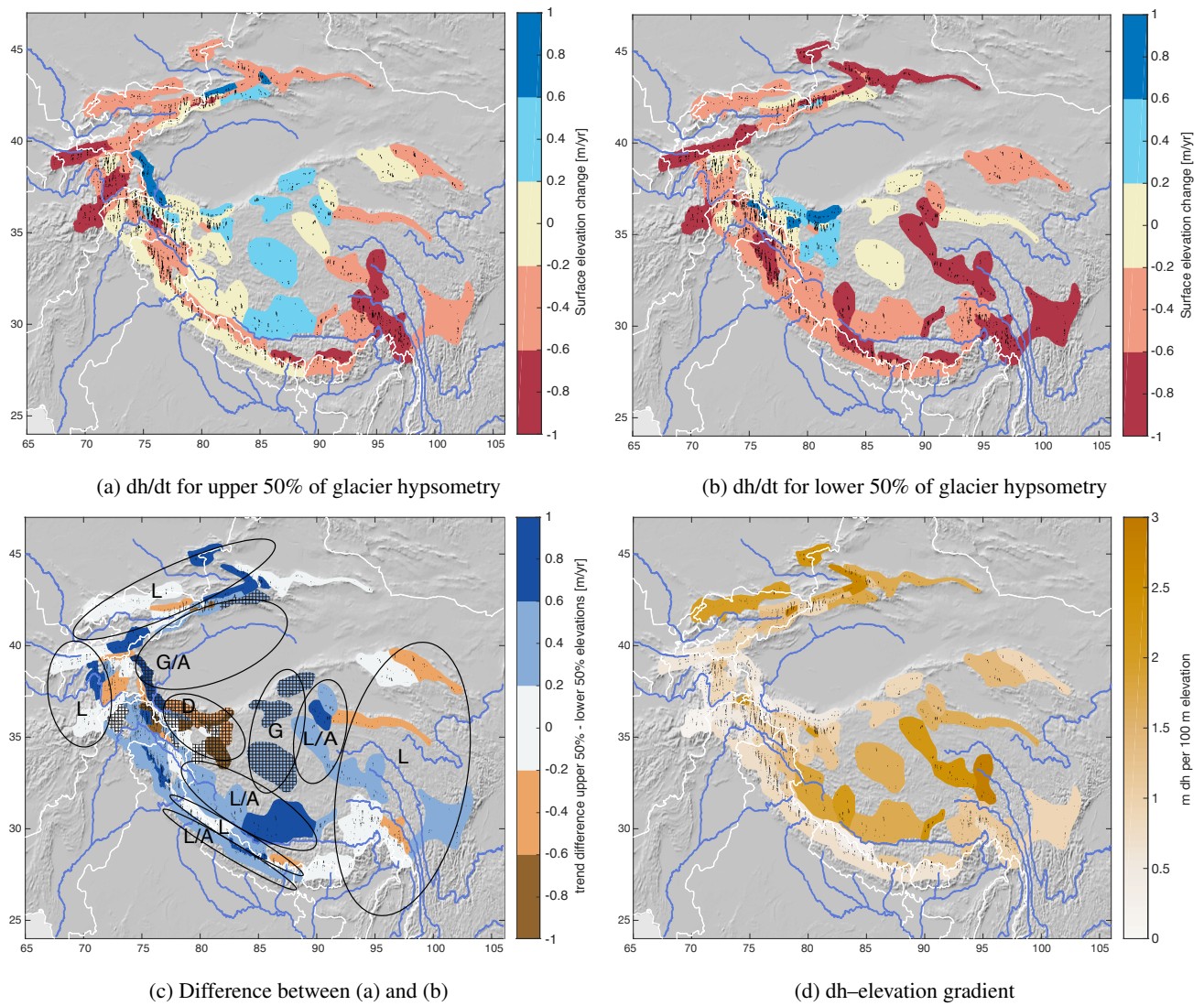

(a) dh/dt for upper 50% of glacier hypsometry

(b) dh/dt for lower 50% of glacier hypsometry

(c) Difference between (a) and (b)

(d) dh–elevation gradient

**Figure 3.** Glacier accumulation and ablation areas indicate regionally different, distinct glacier evolution: glacier surface elevation change for (a) upper 50% and (b) lower 50% of glacier hypsometry, and (c) the difference of the two (upper minus lower). Shaded: areas with overall thickening; L – thickness loss on entire glacier, G – thickness gain, A – adjusting glacier geometry with thinning ablation areas, D – dynamic adjusting of glacier geometry with thickening ablation areas. (d) Gradients of dh (ICESat–SRTM surface elevation) with elevation. Steep dh–elevation gradients may be caused by high SRTM penetration depths in dry, cold accumulation areas and/or from glaciers adjusting their geometry.

## 4.2  Biasing influences from dh–elevation gradient and December 2008 snow fall

The dh–elevation gradients are in some units very steep (Fig. 3d). This means that the surface elevation differences between ICESat and the SRTM DEM are very negative on glacier tongues but very small or even strongly positive in the upper accu-

mulation areas. Steep dh–elevation gradients can result from altitudinal dependency of radar penetration or glacier geometry changes between SRTM and ICESat surface elevation acquisitions. The steeper the dh–elevation gradients are, the stronger is the biasing influence from glacier hypsometry missampling. On the TP and in the northern and eastern ranges of the Tien Shan, the gradients range between 1.5–2.5 m per 100 m elevation. Glaciers in these areas typically occur within an elevation range of ca. 1000 m. In the Nyainqêntanglha Shan/Hengduan Shan, West Kunlun Shan, Karakoram, southwestern Tien Shan and the highest Pamir mountains, dh–elevation gradient values are 1–1.5 m per 100 m elevation. The gradients are moderate ($< 1$ m per 100 m elevation) in the Himalaya, East Kunlun Shan, lower in Pamir, and lowest in the Hindu Kush (0.14 m per 100 m). Our method ensures that any bias from inconsistent sampling of glacier elevations for individual ICESat campaigns is corrected. Neglecting the effect of glacier hypsometry missampling or a trend in sampled glacier elevations would result in considerable bias: on average $\pm 0.13$ m a$^{-1}$, but up to $> \pm 0.3$ m a$^{-1}$ for three units each in Tien Shan and Karakoram.

Correcting dh retrieved from the December 2008 campaign for the effect of increasing snow cover has an unexpectedly large influence on glacier surface elevation change rates (Supplementary Fig. S1c). Elevation changes from corrected dh are on average 0.088 m a$^{-1}$ more negative/less positive. The maximum effect of the December 2008 correction is as much as $-0.25$ m a$^{-1}$ (in unit N2; for off-glacier samples: $-0.11$ m a$^{-1}$ in unit H2), which is a considerable difference given that it is caused by only ca. 10% of all samples (half of one of five campaigns). The potential biasing effect is in fact greatest in areas where MERRA-2 data suggests snow fall during October/November/December 2008 and where off-glacier samples suggest a positive surface change trend (Supplementary Information S1). However, in 20 out of 100 units we were not able to compute the potential biasing effect of December 2008 snow cover (e.g., due to lack of off-glacier samples). To ensure a consistent approach, we did therefore not apply this correction to the results presented above but instead added the difference due to bias correction to the error budget (Fig. 2c). The corrected glacier surface elevation change rates are included in the data supplement. A discussion of the effect of this and other corrections and biasing influences is provided in Appendix D.

## 4.3 Lake changes on the TP

We receive valid (according to our filter procedures) water volume change time series for 89% of the median lake area (74% of all endorheic lakes) on the TP: 1009 lakes with SRTM-based lake surface elevations, thereof 103 also having ICESat-based lake surface elevations (59% of the lake area). Extrapolated lake levels based on annual or campaign ICESat data (Appendix C) yield the same results, but ICESat-based lake level change is on average 1.55 times larger than SRTM-based values. Likely, the reason for this difference is the greater uncertainty of SRTM DEM elevations and pre-2000 SRTM lake levels (Appendix C1). Multiplied with areal changes to receive volume changes, the relative difference is reduced to 1.09 times. Average 1990–2015 water level increase corresponds to 0.14 m a$^{-1}$ (SRTM) and 0.18 m a$^{-1}$ (ICESat) in lake-level change per year (Fig. 4a, robust linear regression of dV). All, except a handful of lakes predominantly in the very south of the TP, grew during the studied time period, and growth of individual lakes is largest in the northern and eastern part of the TP. Figure 5 shows relative lake volume growth (based on SRTM lake levels) for individual lakes and regional medians over time for six regions: southwestern (SW), eastern (E), central (C), northeastern (NE), northwestern (NW) TP and Qaidam Basin/Qilian Shan (QQ), indicated in Fig. 4a. (Note that Fig. 5 shows the timing of the volume change and not changes in absolute lake volumes, these are unknown). Rather

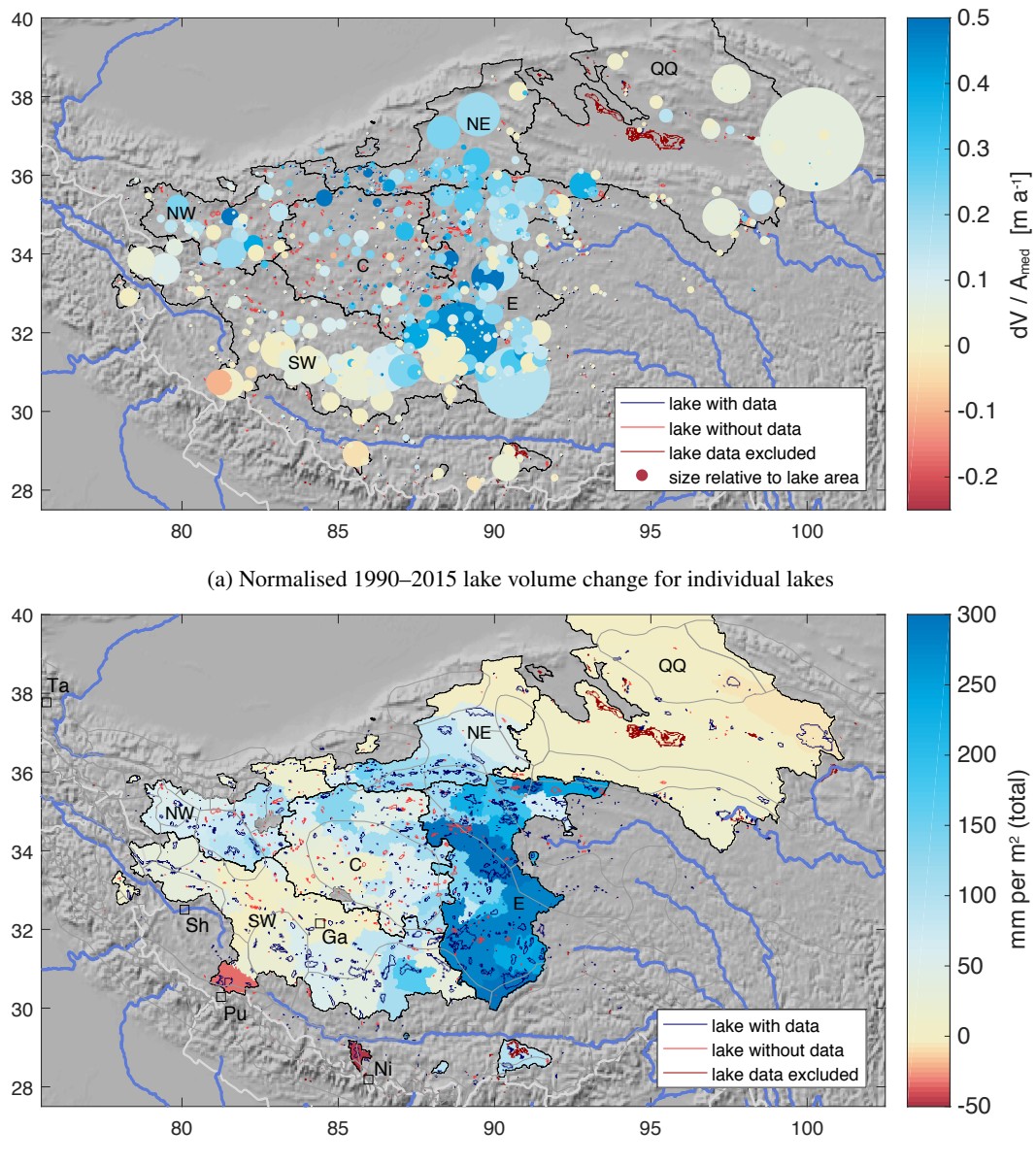

(a) Normalised 1990–2015 lake volume change for individual lakes

(b) Annual specific water change per endorheic catchment between 1990–1999 and 2000–2009

**Figure 4.** Lake volume changes on the Tibetan Plateau. (a) Normalised lake volume change for individual lakes. Colours show the average annual 1990–2015 lake level change in metres (volume changes dV divided by median lake areas to receive comparable values for lakes of different sizes). Circle areas are scaled relative to lake areas. (b) Specific water change per endorheic catchment for the decadal difference between 1990–1999 and 2000–2009 lake volumes. Values correspond to the sum of individual lake water volume changes (average changes assumed for lakes with missing data) divided by catchment area to make their units comparable to precipitation sums. Red lake outlines: lacking plausible data; purple lake outlines: lakes excluded due to human influence on lake levels/extent. Squares: meteorological stations. Labelled regions with black outlines are referred to in the text.

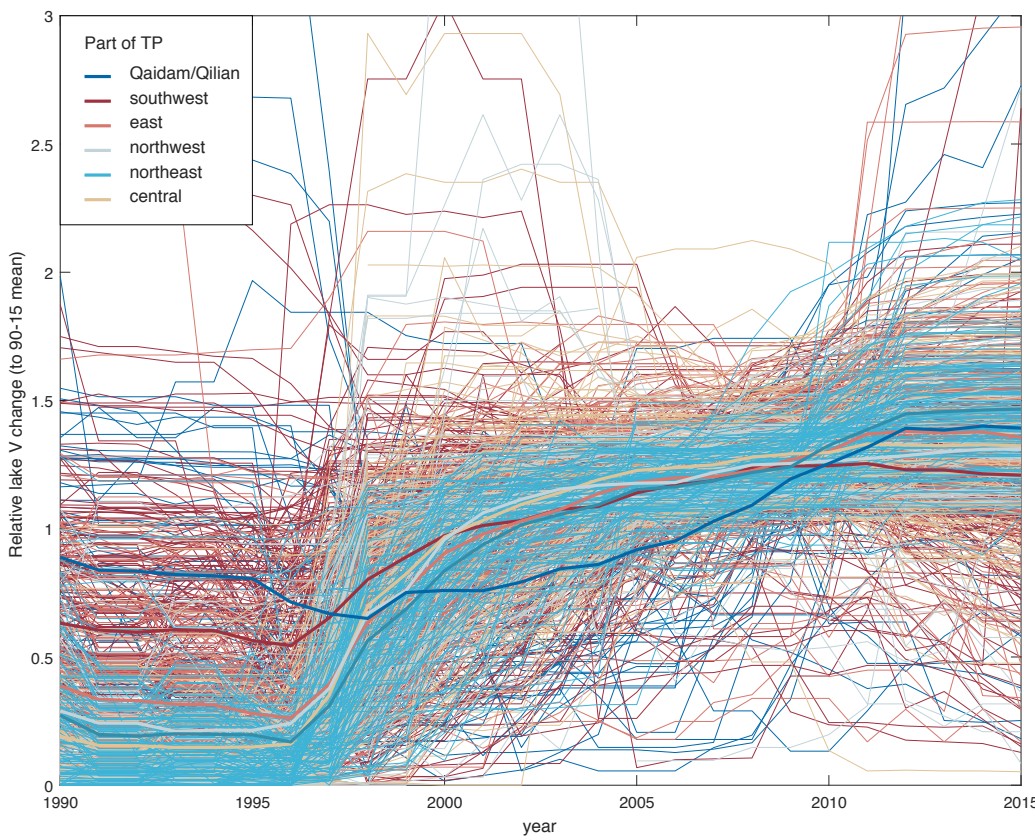

**Figure 5.** Relative lake volume change for individual lakes on the Tibetan Plateau, coloured by region. Volume changes dV are normalised by the 1990–2015 mean dV for comparability, annual values are median-filtered (7 years window size). Thick lines indicate the median for each region. The regions northeast, northwest and central correspond to areas with observed 2003–2008 glacier thickening.

than growing steadily, most lakes seem to have undergone a phase of sudden and rapid growth starting between 1995 and 2000, and gradually slowing down until ∼2009, with rather stable conditions before and after this period. (Note that lake time series are median-filtered due to data scarcity for the years 1995–1999. There is thus some uncertainty on the exact timing of the onset of lake growth.) Relative lake volume change was most sudden and rapid for the northeastern, northwestern, central and eastern
5  TP (the former three corresponding to areas with 2003–2008 glacier thickening). Lakes in the southern and southwestern part of the TP showed more varying and overall less growth, with a tendency to decrease after 2010. Endorheic lakes in the Qaidam basin/Qilian Shan region further northeast also show a different and more varying evolution with slower growth that started only around 2004, but continued until ∼2012. The latter effect is also visible for the adjacent lakes on the northeastern TP (east Kunlun Shan).
10    Figure 4b shows the corresponding specific water volume change per endorheic catchment as the decadal difference between 1990–1999 and 2000–2009 average lake volumes (based on SRTM lake levels). In other words, the figure shows the accumu-

**Table 1.** Precipitation/lake water volume changes between decadal averages of the 1990s and the 2000s per basin region (dP and dV), and annual glacier mass balance of adjacent glacierised areas for 2003–2008 (last column). dV: total decadal lake water volume difference in mm m$^{-2}$, dP: annual precipitation difference in mm m$^{-2}$a$^{-1}$, station order in southwest TP: Shiquanhe, Gaize, Pulan, Nielaer. Glacier surface elevation changes are converted to mm w.eq. a$^{-1}$ assuming a density of 850 kg m$^{-3}$.

| Region | dV SRTM | dV ICESat | dP MERRA-2 | dP ERA Interim | dP stations | Glacier mass balance |
|---|---|---|---|---|---|---|
| Southwest TP | 39±11 | 59±16 | 81±33 | 15 ± 31 | −1 ± 14, 42±17, 19 ± 16, 60±50 | −33 ± 11 to −10 ± 14 |
| East TP | 252±33 | 275±37 | 100±18 | 30 ± 14 | | −17 ± 10 to −8 ± 14 |
| Central TP | 69±10 | 71±11 | 56±5 | 25 ± 8 | | 21 ± 38 |
| Northwest TP | 62±14 | 70±15 | 34±11 | −33 ± 11 | 16 ± 72 | 29 ± 10 to 31 ± 9 |
| Northeast TP | 60±12 | 54±9 | 85±13 | −2 ± 22 | | 13 ± 11 to 50 ± 21 |
| Qaidam / Qilian | 1±5 | 1±4 | 87±14 | 24 ± 17 | | −25 ± 14 to −13 ± 10 |

lated additional water volumes evenly spread across the entire catchment areas. The pattern of predominant water volume increase especially in the northern and eastern TP compares well to the results in Fig. 4a. Lake volume growth on the eastern TP is accentuated due to considerably larger lake areas and lake density compared to the mostly small lakes further north/west. Table 1 shows the additional water volumes accumulated between the two decades as in Fig. 4a for the same regions as above (corresponding mass changes in Gt are provided in Supplementary Information S2). To yield values comparable to precipitation changes, the reader has to divide the total decadal differences dV given in the table by the number of years during which the additional water was accumulated. Assuming the change happened rather gradual during the entire decade, the specific annual water change would correspond to 1/10 of the values in Table 1. For instance, for water volumes using SRTM-based lake levels: 25±3 mm a$^{-1}$ for the eastern TP, 4±1 mm a$^{-1}$ for the southwestern TP, 6–7±1 mm a$^{-1}$ for the central and northern TP, and 0.1±0.5 mm a$^{-1}$ for the Qaidam basin/Qilian Shan region. Notably, there are considerable differences between catchments within each region (range for SRTM-based estimates: –5±1 to +35±6 mm a$^{-1}$, excluding one outlier of 163±7 mm a$^{-1}$ for the catchment centred at 34.3° N / 88.8° E). The estimates based on SRTM and ICESat lake levels aggregated for the six regions nevertheless agree very closely. The above annual values have to be doubled, or the dV values given in the table multiplied by 1/5, for instance, if one prefers to assume that the water volume increase happened during 5 years only, with stable conditions before and after — an assumption which also is plausible from Fig. 5.

## 4.4 Precipitation increase on the TP

A change in precipitation could explain both lake growth and glacier mass balance (if dominated by precipitation rather than temperature/melt). When subtracting the part that is lost through evaporation, the precipitation change should yield numbers

that are directly scalable in relation to glacier mass balance and endorheic catchment water volume (when neglecting changes in subsurface water transport).

Annual precipitation sums on the TP from meteorological stations range from as little as 50–100 mm $a^{-1}$ (Shiquanhe and Tashkurgan stations, southwest TP and West Kunlun Shan) to 500–900 mm $a^{-1}$ (Nielaer station, southern TP). Reanalysis
values of both products used, MERRA-2 and ERA Interim, lie in between. All datasets record the majority of precipitation (>70%) during the monsoon-influenced summer months (May–September), except for Pulan and Nielaer, the two southern-most stations close to the Himalaya (only ca. 50% precipitation in summer). On the data-sparse TP, both station data and reanalysis products may contain bias due to the stations not being representative for a larger area and the lack of observational forcing data for reanalysis products, respectively. We thus use the data in a summarised way and focus on relative changes
rather than relying on absolute numbers to detect/confirm temporal changes and large-scale spatial patterns.

Of the five meteorological stations available, especially Shiquanhe and Pulan show little change in precipitation and pan evaporation (Fig. 6). The Gaize station, located most central on the TP but still more south than our corresponding glacier unit, indicates a stepwise precipitation increase around the year 2000, but data from only one station need of course to be interpreted with care due to potential local effects and changes to the station. A more gradual increase is visible in the Tashkurgan data.
Differences in decadal average precipitation range from −1 (Shiqanhe) to +60 mm (Nielaer) within 10 years, notably with greatest relative change for the Gaize station (+42 mm per decade, a 25% increase) and Tashkurgan station (+16 mm or +22% per decade). Decadal differences are mostly (Nielaer, Tashkurgan) or exclusively (other stations) caused by an increase of precipitation during summer months. Pan evaporation reaches twice to tenfold of precipitation sums.

The two reanalyses used here differ considerably both in precipitation evolution and in estimated evaporation (Fig. 7) and
also the spatial patterns of precipitation changes differ (Fig. 8). Figures 7a (ERA Interim) and 7b (MERRA-2) show regional averaged annual sums for total precipitation, evaporation, and the difference of the two, for grid points within the TP lake catchment regions defined above. Notably, ERA Interim suggests considerably higher evaporation values than MERRA-2, in particular for the southwestern TP (SW) and the three northern regions (NE, NW, QQ), resulting in much lower suggested net water availability in the areas where we see glacier thickening than it is the case for MERRA-2. Both reanalyses show
an increase in precipitation starting from ca. 1995, but for ERA Interim, the evolution only lasts until ca. 2000 after which precipitation sums decrease. Also, the short-term precipitation increase is not visible for the northern parts of the TP. Fig. 8a shows the spatial distribution of the decadal difference between average summer precipitation in 1990–1999 vs. 2000–2009. ERA Interim data suggests only a marginal precipitation increase on the TP and a considerable decrease in decadal average precipitation for the Kunlun Shan area (−33±11 mm for northwestern TP, table 1).
MERRA-2, on the other hand, rather suggests a stepwise precipitation increase (Fig. 7b) with continuously higher precipitation sums until ca. 2010 for the entire TP, and even a continuous increase through 2015 for the northern part of the TP. For all six regions, this results in a total increase in precipitation of 34±11 mm (northwestern TP, table 1) to 100±18 mm (eastern TP) mm within 10 years. Except for the Qilian Shan region, the change is exclusively driven by increasing summer precipitation. Winter precipitation did not change noticeably (−9 to −2 mm decadal change for the five TP regions, +8 mm for Qilian Shan).
Fig. 8b shows the spatial distribution of summer precipitation change (difference between decadal averages). Compared to the

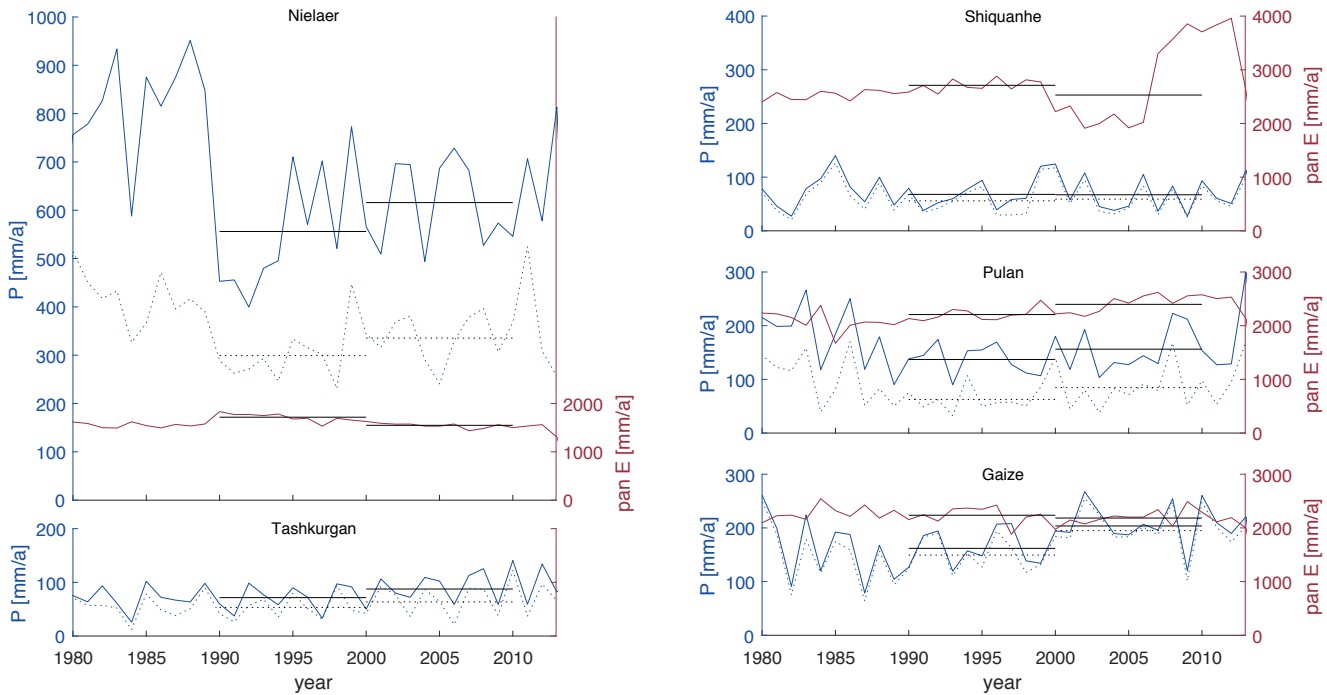

**Figure 6.** Annual precipitation (P, blue line), pan evaporation (pan E, red line), and summer precipitation (dotted lines) for five stations on the southern and western TP (Fig. 4b).

same map with ERA Interim data, MERRA-2 suggests a considerably stronger precipitation increase on the TP and increasing precipitation also in the Kunlun Shan area. For both reanalyses, the spatial patterns are the same for annual precipitation rather than summer precipitation only (not shown).

The two reanalysis products agree somewhat better when precipitation numbers are corrected with estimates of actual evap-

5 oration to assess the total decadal increase in water availability. For MERRA-2, the decadal difference is then reduced to $6\pm11$ mm (central TP) to $68\pm13$ mm (northeastern TP). However, the evaporation-corrected increase is greater when looking at summer months only ($31\pm7$ mm to $77\pm11$ mm per decade, compared to a decrease in water availability during winter months of $-27\pm4$ mm to $-6\pm3$ mm, not shown). Corresponding ERA Interim increase in annual water availability is $14\pm32$ to $38\pm12$ mm (summer: $-19\pm11$ mm in the Kunlun area to $38\pm13$ mm, winter $-16\pm6$ to $5\pm4$ mm). Both datasets suggest that 30–

10 60% (MERRA-2) or 13–50% (ERA Interim) of precipitation on the TP falls as snow during the summer months and that the proportion of snow fall did not change noticeably between the decades (not shown).

The regions where MERRA-2 indicates increased summer precipitation correspond well with those areas on the TP and in Eastern Kunlun Shan with moderately negative to positive surface elevation change and/or endorheic lake growth. ERA Interim data indicates a similar pattern but the lower (TP) precipitation increase and, particularly, the decrease for the Kunlun Shan

15 region (Fig. 8a) does not fit well with the results from our lake and glacier data.

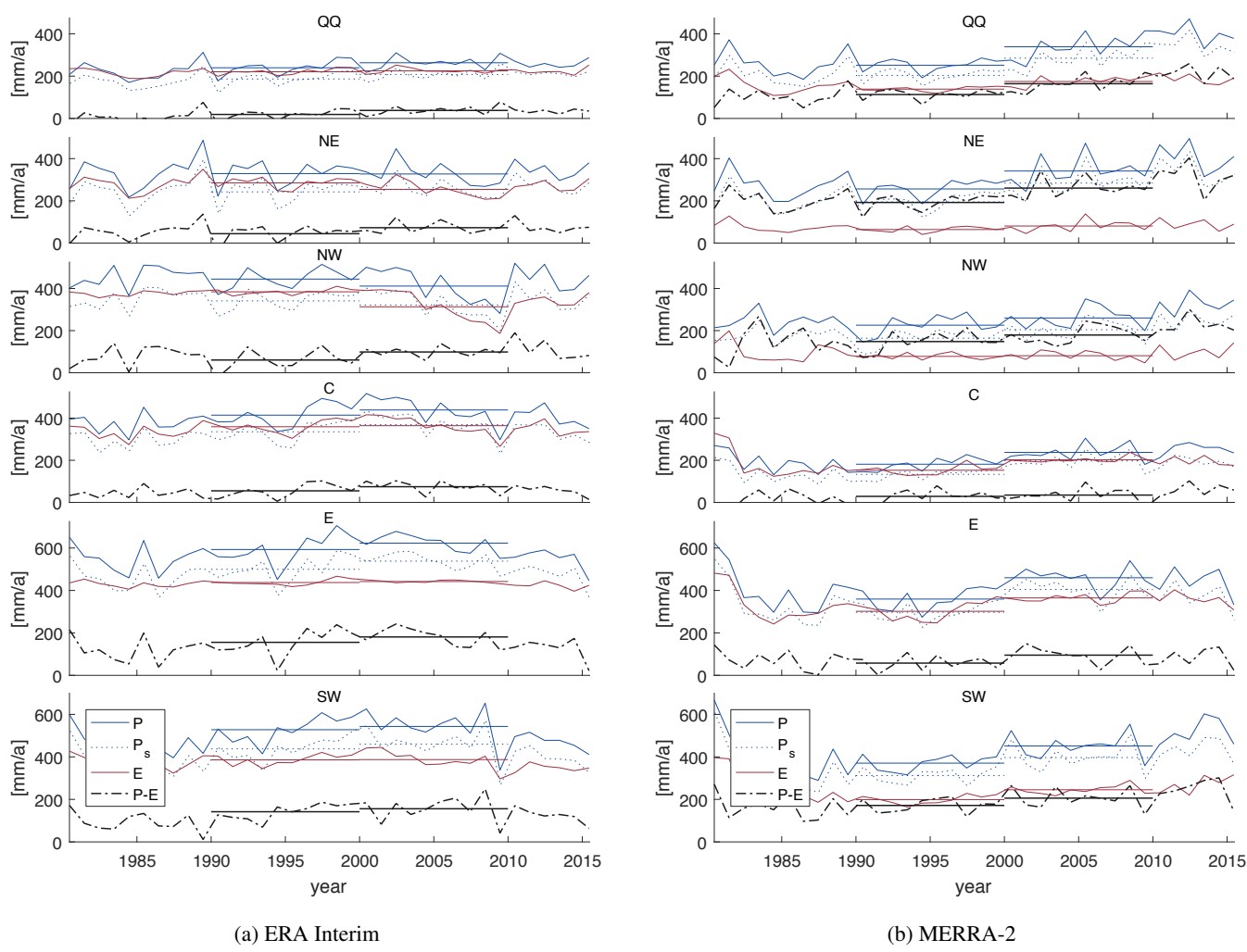

(a) ERA Interim

(b) MERRA-2

**Figure 7.** Timeseries of annual total precipitation (P), evaporation (E), the difference of the two (P-E), summer precipitation (Ps, May–Sept), and their respective decadal averages, for reanalysis grid points within the six lake change regions on the TP: southwestern (SW), eastern (E), central (C), northwestern (NW), northeastern (NE) TP and Qaidam Basin / Qilian Shan (QQ).

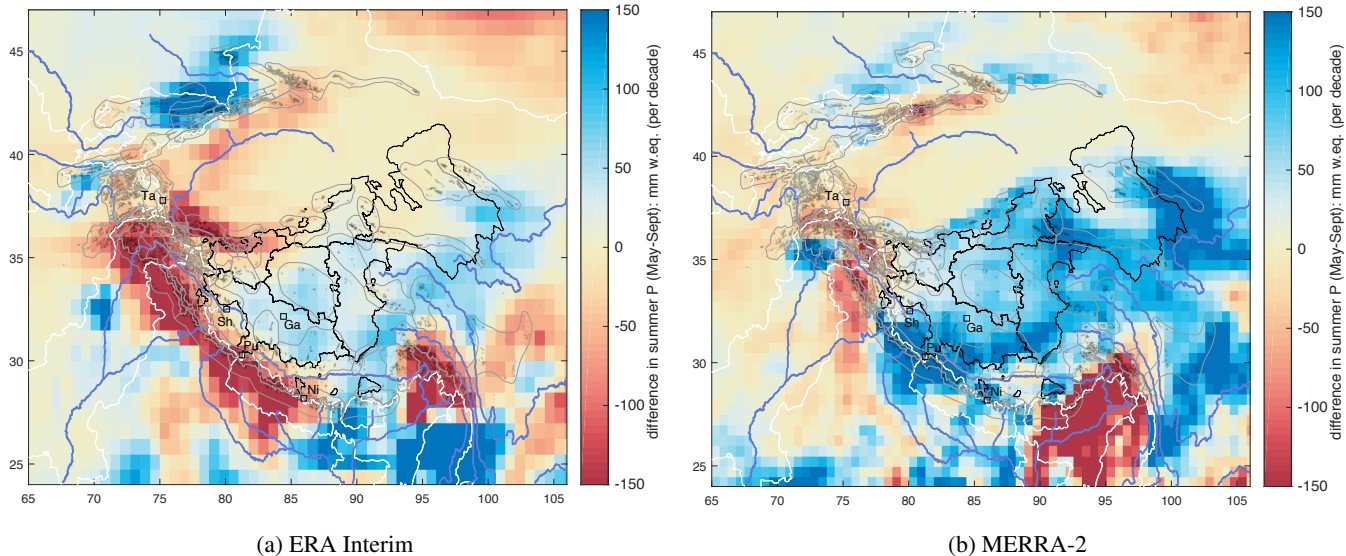

**Figure 8.** Difference between decadal averages of summer precipitation (May–September) in 2000–2009 and 1990–1999. MERRA-2 suggests a considerable increase on the TP and in the Kunlun Shan, compared to a negligible increase on the TP and decreasing precipitation in the Kunlun Shan for ERA Interim.

Our above results on precipitation changes relate to decadal means in order to enable systematic comparison to other data. It is however important to note that these results vary if other time periods are chosen for aggregation. Kääb et al. (2018), for instance, summarize total annual precipitation amounts estimated from ERA-interim reanalysis over the Aru region, nortwestern TP, over 1979–1995 and 1995–2008 to suggest a 33% increase between the periods.

## 5  Discussion

The 2003–2008 ICESat surface elevation changes paint a spatially diverse picture of glacier changes in HMA. The general pattern — glacier volume gain in the Kunlun Shan and the inner TP and glacier volume loss elsewhere — appears robust, no matter whether we aggregate the samples in a regular grid or manually delineated units. The more distinct spatial pattern agrees with the ICESat studies of Kääb et al. (2015, 2012), the ASTER-based geodetic mass balances of Brun et al. (2017) and with the overall picture drawn by the previous regional studies of Neckel et al. (2014), Gardner et al. (2013) and Farinotti et al. (2015) based on data from ICESat, GRACE and modelling. The pattern found is also robust against small changes in reference elevations (such as from using the 1 arc-second SRTM DEM) or sample composition, and can also be reproduced using the most recent RGI glacier outlines — which have clearly become much more accurate since the study of Gardner et al. (2013).

On a local scale, and in contrast to the above regional view, there are considerable differences to previous findings in glacier changes, including the ones based on the same ICESat data. Compared to a visualisation of our results in a regular grid, we find that spatial aggregation matters: even within our study, only the manual zonation brings forward finer spatial differences

e.g. from topographic-orographic setting. Our results also suggest that inconsistent sampling hypsometry, snow cover, and local vertical biases and elevation inconsistencies can have a severe biasing effect on ICESat-based glacier changes when not accounted for properly — in particular where they vary for different ICESat campaigns. A method discussion, in particular on biasing influences on ICESat glacier surface elevation change rates, is provided in Appendix D.

## 5.1 Coincident lake growth and glacier thickening

The regions with glacier thickening, or thickening of upper glacier areas (Fig. 3a), spatially match the areas with growing endorheic lakes on the TP and where MERRA-2 data suggests a stepwise increase of summer precipitation around the year 2000. The change in available precipitation amounts, lake water volume, and glacier mass balances are of the same magnitude and match well in terms of timing. Studies analysing individual lake time series suggest the increase started closer to the year 2000 (Lei et al., 2013; Zhang et al., 2017, 2018; Song et al., 2015) than our Fig. 5 suggests. This could be due to the application of a median filter which contributes to shifting the onset of volume change in the middle of a period with large Landsat data gaps (1996, 1997, see App. A4). The recent growth of TP's lakes is established by numerous recent studies (e.g., Zhang et al., 2011, 2013; Song et al., 2015; Zhang et al., 2017). In this study, lake volume changes on the TP serve as proxies for precipitation changes, but they may also help resolving satellite gravimetric signals to compute glacier mass changes (see introduction). The fact that glacier volumes are increasing in regions where also lake volumes increase, and the fact that lake volumes are also increasing in little or not glacierised basins, both suggest that the increases in lake volumes over the study region are not mainly driven by increased water influx from glacier mass loss (see e.g. Song et al., 2015).

Though, glacier mass loss can certainly play an additional role for lakes with declining glaciers in their catchment. This is in line with, and extends geographically, water balance modelling by Lei et al. (2013) for six selected lakes in our East TP zone (Fig. 4b) that suggests mainly precipitation increases to be behind the increases of lake volumes, accompanied by decreases in potential evaporation due to decreasing wind speed, and to a lesser extent increase in glacier runoff (Song et al., 2015). Evaporation may also have decreased due to increased humidity from higher precipitation amounts. For 1981–2013, Zhang et al. (2018) find a significant decrease of pan evaporation from meteorological stations on the Eastern TP (these are however further east than the endorheic lakes). For the Siling Co lake in our East TP region, potential evaporation showed stable conditions or a slight increase between the mid/end 1990s to 2010 although it was decreasing overall over 1961–2010 (Guo et al., 2019), underlining even more the key role of precipitation increases for the observed lake volume increase. The reanalysis products used in this study do not show a coherent signal for evaporation. They suggest relatively stable (ERA Interim) or increasing (MERRA-2) evaporation in the southern three regions of the TP, and decreasing (ERA Interim) or roughly stable (MERRA-2) evaporation for the two regions in the north. It is noteworthy that correcting precipitation data with evaporation allows to somewhat reconcile the two reanalysis datasets: Also ERA Interim shows an increase in so-computed net water availability, although it is smaller than for MERRA-2.

Lei et al. (2013) suggest that groundwater exchange between different basins has very limited influence on the water balance of each lake due to the impermeability of surrounding permafrost. Such groundwater exchange does not affect the basin-wide water volume changes of this study, but thawing permafrost could be another potential source of water. An increase of

active layer depth also causes an increase in groundwater storage capacity in ice-free ground an may change the amount of precipitation or water from snow melt that is retained or released (S. Westermann, pers. comm.). However, we are not aware of studies that quantify the amount of water available from these processes. Modelling studies (Ran et al., 2018; Zou et al., 2017) find continuous permafrost in the northern part of the TP (our regions NW, NE, C and most of E) and discontinuous permafrost

including larger areas of non-frozen ground in the southern/eastern parts of the TP (our regions SW and most of QQ). Recent and ongoing temperature rise led to an increase in the active layer and degrading permafrost that seems to have been greatest during the 60s and 00s and in the southern and eastern parts of the TP (Ran et al., 2018), where we find little lake change (SW) and strong lake growth (E), respectively.

## 5.2    Precipitation increase on the TP and glacier sensitivity to these changes

In particular the MERRA-2 reanalysis, and to a lesser degree also ERA Interim and station data, suggest precipitation on the TP has increased around 1995–2000. The spatial patterns of decadal precipitation increases and the glacier growth on the TP and in the Kunlun Shan suggest a causal relationship. Increased precipitation in the region has been noted before: Yao et al. (2012) attributed a pattern of precipitation/glacier changes to a strengthening of the Westerlies while the Indian monsoon is weakening. A rise in extreme precipitation events at stations in the study region was attributed to a weakening East Asian monsoon (Sun

and Zhang, 2017). Fujita and Nuimura (2011) and Sakai and Fujita (2017) model a decrease in theoretical equilibrium line altitudes (ELA) in western Tibet between 1988 and 2007, and attribute these trends to increasing precipitation in western Tibet (but decreasing precipitation in western Pamir and the western Himalaya). Glaciers in West Kunlun were in general shrinking between 1970 and 2001, only those on the south slope were already growing between 1991 and 2001 (Shangguan et al., 2007).

While the reanalysis data does not suggest an increase in summer precipitation in Eastern Pamir and on the western and

northern boundary of the Tarim Basin, Tao et al. (2011, 2014) found indications for a wetter climate and increasing streamflow in the entire basin. Shi et al. (2007) suggest that a shift from a warm-dry to a warm-wet climate in the entire northwest of China happened already around 1987. Our results indicate that glaciers on the southernmost orographic barrier in the Tien Shan are closer to balance than glaciers further north/west. We thus speculate that the change in circulation patterns behind the positive precipitation change, centred further south, extends across the entire Tarim basin, and with it more favourable conditions for

glaciers on the edge of the entire basin.

Lack of meteorological observations on large parts of HMA result in substantial uncertainties with recent precipitation changes on the TP (Kang et al., 2010) and available gridded precipitation datasets (Sun et al., 2018; Smith and Bookhagen, 2018). While they are also affected by the lack of direct observations, reanalysis products are an important source of physically based model data in such data-sparse regions (Cuo and Zhang, 2017). Orsolini et al. (2019) find that MERRA-2 does not model

snow depth or snow cover fraction well on the TP, but still best matches total precipitation amounts on the TP compared to ERA Interim and other reanalysis products which overestimate precipitation compared to reference data. Assimilation of snow observations and a better parametrisation of snow-related physical processes are thus needed to improve model performance for the often thin and short-lived snow cover should improve future reanalysis products on the TP (Orsolini et al., 2019). Given the importance of evaporation in this dry region and how much the two analysed reanalysis products differ in this regard, it

seems also evaporation could be better represented in the models. Improved spatial resolution should contribute to better model high-altitude precipitation due to the importance of spatial resolution to capture orographic processes. Examples are the High Asia Reanalysis HAR (Maussion et al., 2014, available for most of HMA but unfortunately a time span of 10 years only) or the upcoming ERA5 Land reanalysis which is more suitable for mountainous areas than ERA5 (Orsolini et al., 2019).

Maussion et al. (2014) proposed a new classification for HMA glaciers based on their main accumulation season from 2000–2011 HAR precipitation data. Our pattern of positive glacier changes matches very well with their classification of the predominant glacier accumulation season as spring or early summer. On the TP, Maussion et al. (2014) find a gradual transition towards later accumulation (monsoon-dominated) whereas there is a crisp boundary to winter accumulation in the Karakoram/Pamir. Both patterns correspond to the zonal boundary of 'extreme continental (polar) glaciers' suggested by Shi and Liu (2000),
which encompasses the northwestern half of the TP, glaciers north of central Karakoram, the easternmost Pamir, and the entire Kunlun Shan. On a coarser spatial and longer temporal scale, Kapnick et al. (2014) suggest that glacier accumulation in the Karakoram is least sensitive to atmospheric warming due to dominating non-monsoonal winter precipitation in this region.

Forsythe et al. (2017) attribute summer cooling in the Karakoram since the 1960s to a southerly shift of a circulation system that they named Karakoram vortex. In the Karakoram area the southerly shift leads to increased passage of westerly depressions
and corresponding cooler temperatures due to increased cloud cover and decreased insolation. The effect of this may extend to the areas to the north, namely the Kunlun Shan, Pamir, Tien Shan and Tarim basin (see their Fig. 2b). De Kok et al. (2018) model the effect of increased irrigation intensity in the lowlands of HMA and find that they may cause increased summer snow fall and a decrease in net radiance in the Kunlun Shan and parts of Pamir and northern Tibet.

Fujita (2008) finds that HMA's glaciers are more affected by precipitation seasonality and concentration than by changes in
annual precipitation. Where accumulation and warming happen at the same time (i.e. summer), rising temperatures increase both melt and the share of precipitation that falls as rain instead of snow. While temperatures are rising in entire HMA, the glacier sensitivity study of Fujita and Nuimura (2011) suggests that temperature was not the limiting factor for glacier existence everywhere. In the extremely dry and cold TP and Kunlun Shan, with glaciers and in particular their accumulation areas at high elevations (Fig. 1), glacier growth due to increased precipitation is thus entirely plausible — despite a warming trend.
This also stresses that the elevation of HMA glaciers (Fig. 1) is an important factor in their respective responses to temperature and precipitation changes (Sakai and Fujita, 2017), and thus in the here-observed glacier volume changes.

## 5.3  Glacier geometry changes on the TP

In light of continued climatic changes and rising temperatures in the study region, ICESat only provides a short snapshot of ongoing glacier reactions. This snapshot falls exactly into the decade where an increase in precipitation on the TP around
the year 2000 would cause the largest effects on glacier volume changes: with some delay, glaciers dynamically change the geometry of their ablation areas (which are thickening) to adjust to a new glacier equilibrium state (Kääb et al., 2018; Gilbert et al., 2018). Ke et al. (2015) and Bao et al. (2015) report such stronger surface elevation gain for ablation areas compared to elevation gains in accumulation areas in what they refer to as West Kunlun Shan (our unit KS1, plus four to the North and East of it). As visible in Fig. 3c, we find the same signal for a larger area of an additional eight adjacent units, including those to the

South (area marked "D"). Care has to be taken when analysing elevation changes over only parts of a glacier as this violates the condition of mass continuity. Thickening of the ablation parts of a glacier can be caused by either positive surface mass balance or dynamical changes (i.e. increased ice flux). In the case of West Kunlun Shan, a stronger thickening of the tongues compared to upper glacier areas could indicate that both were happening: a general glacier thickening from ongoing positive mass balances, plus a delayed dynamical thickening from earlier mass gain in the accumulation areas.

The rate of warming on the TP is greatest for the elevations where glaciers have their ablation areas (Yao et al., 2012; Ran et al., 2018). In the southeastern part of the TP, dh–elevation gradients are largest (darker units in Fig. 3d), which could indicate that dynamical changes are happening also there: an overall thinning signal could be composed of increased melt at lower elevations, causing strongly negative dh, while the accumulation areas are thickening or stable due to increased precipitation/accumulation, causing stable surface elevations or positive dh. This interpretation is supported by the gradual transition visible in Fig. 3c: in East Kunlun Shan and central TP, we see a thickening of accumulation areas and no change on the tongues (area marked "G"), and further east/south accumulation areas experienced little change but tongues were thinning (marked "L/A").

Dynamic glacier geometry adjustments might also be reflected in glacier flow. Dehecq et al. (2019) found that for the 2000–2016 period, the flow speed of HMA glacier tongues decreased everywhere but in the Kunlun Shan and Karakoram and only slightly decreased on the TP. While the different time periods and spatial aggregation don't allow a more detailed comparison, their results confirm that these regions were not or less affected from rapid glacier mass loss with thinning and increasingly inactive tongues.

### 5.4 Glacier thinning on the Eastern Tibetan Plateau

The negative elevation change rates on the eastern border of the TP agree with reported glacier mass loss in this area, although varying annually and in space (Kang et al., 2009; Yao et al., 2012). For this part of the southeastern TP, Mölg et al. (2014) found that the competition between the monsoon and large-scale westerly waves of the mid-latitude circulation in spring/early summer determines annual mass balance. The south–north transition of the jet stream across the TP in spring varies in timing and efficiency, and its re-intensification in summer on the northern edge of the TP is related to the onset of the summer monsoon (Schiemann et al., 2009). This interplay affects both precipitation and summer air temperature. All glaciers in the region are of summer accumulation type, except for East Nyainqêntanglha Shan and Hengduan Shan (Maussion et al., 2014). The area where the atmospheric flow strength over the TP correlates strongly with summer temperatures (Mölg et al., 2014) forms an arc-shaped band from the above mentioned mountain ranges along the northern slopes of the East Nyainqêntanglha Shan to the easternmost glacierised mountains in the area. The correlation of Monsoon/Westerlies competition with temperature is decreasing rapidly north towards the easternmost Kunlun Shan and south to the Goikarla Rigyu range just north of the Yarlung Tsangpo Valley. This pattern corresponds well with our findings of only slight glacier thinning in Goikarla Rigyu/East Kunlun Shan (units N1 and KS2) but more negative volume changes in the easternmost HMA glaciers (our unit HS). Reconstructed mass balances from six glaciers on the eastern slope of Minya Gongga (in the very east of unit HS) were $-0.79 \, \text{m w.e. a}^{-1}$ in 2001–2009, a notable further decrease from an already negative average of $-0.35 \, \text{m w.e. a}^{-1}$ in 1952–2000 (Zhang et al.,

2012). Converted to mass loss, our results in this area are $-0.75 \pm 0.43$ m w.e. a$^{-1}$ — the large uncertainty reflects the sparse glacier coverage and low sample numbers in this unit. Zhang et al. (2012) report that both the ELA and temperatures in the beginning and end of the melt season were strongly rising during the ICESat decade.

Glaciers in the Qilian Shan in the very northeast of the TP have been shrinking less than those further south in the last
decades (Tian et al., 2014). In-situ mass balances on Qiyi glacier were strongly negative in 2005–2006 ($-0.95$ m w.e. a$^{-1}$) but less so in 2006–2007 ($-0.3$ m w.e. a$^{-1}$). The 2006 negative mass balance is indeed visible as a marked decrease between ICESat's 2005 and 2006 autumn campaign median dh in all our units north of Nyainqêntanglha Shan (not shown). We find only moderate thinning in the eastern part of Qilian Shan (converted to mass changes: $-0.26 \pm 0.14$ m w.e. a$^{-1}$), where Qiyi Glacier lies, and even less negative values further west ($-0.14 \pm 0.10$ m w.e. a$^{-1}$), in line with Tian et al. (2014). Towards east,
glaciers become smaller and elevations lower, and the influence of the East Asian Monsoon becomes stronger.

## 5.5 Glacier mass balance and precipitation in the Himalayas

We find consistently less severe glacier thinning on the first orographic ridge across the entire Himalayan Range. Misclassifications of e.g. perennial snow patches with stable surface elevations classified as glaciers would cause a mixed glacier/land trend with a weaker surface lowering signal. To achieve this effect, the misclassification would have to be severe (ca. half of the
samples) and be present in both our manual classification and the RGI, as the pattern is visible with both glacier classifications. We carefully classify our samples manually to avoid precisely such mixed signals, thus we consider this bias unlikely. Another cause could be reduced melt due to insulation from debris cover. It has previously been shown that stagnant (debris-covered) tongues lose mass at a similar rate as clean ice glaciers (Kääb et al., 2012; Gardelle et al., 2012b; Pellicciotti et al., 2015; Ragettli et al., 2016). We thus assume that debris-cover is not the cause of the observed differences.
A potential explanation for the less negative mass balances on the first, and thus wettest, orographic ridge in the Himalaya is a locally lower sensitivity of glacier mass balances to precipitation (and changes thereof). Precipitation from summer monsoon influx decreases sharply after large changes in relief (Bookhagen and Burbank, 2006). Maussion et al. (2014) find that precipitation regimes are strongly varying over short distances in the Himalaya, not least due to glacier orientation on the windward or lee side of the a mountain range. Wagnon et al. (2013) and Sherpa et al. (2016) mention the meteorologically
exposed location of Mera glacier (4949–6420 m a.s.l.) in the Khumbu region, Nepal, as a possible explanation of its roughly stable mass balance since 2007 when in-situ measurements began. This stands in stark contrast to the considerable mass loss seen in Pokalde and Changri Nup glaciers only 30 km further north (the latter are also smaller and located at lower and thus warmer elevations, which likely contributes to these differences). In our ICESat zonation, these glaciers are located in units H1 ($-0.12 \pm 0.25$ m a$^{-1}$) and H2 ($-0.50 \pm 0.32$ m a$^{-1}$). Wagnon et al. (2013) note that in the DEM differencing study of Gardelle
et al. (2013), larger glaciers in the same range as Pokalde/Changri Nup also seem to experience more surface lowering than Mera glacier further south. Our consistently less negative glacier volume changes of the first orographic ridge across the entire Himalayan Range supports the interpretation of Wagnon et al. (2013) and Sherpa et al. (2016), and suggests the effect is visible along the entire Himalayan Range. However, the 2004–2008 average annual mass balances of the well-studied Chorabari and Chhota Shigri glaciers in western Himalaya do not follow this pattern. South-facing Chorobari lies on the outermost orographic

ridge and lost mass at a rate of $-0.73$ m w.e. a$^{-1}$ (Dobhal et al., 2013), which is comparable to north-facing Chhota Shigri's balance of $-0.9$ m w.e. a$^{-1}$ (Ramanathan, 2011). Both glaciers lie at comparable elevations (ca. 4000–6400 m a.s.l.).

The ELA sensitivity study of Fujita and Nuimura (2011) is too coarse to confirm orography-related spatial differences across the Himalaya, but along the mountain ridge their findings correlate well with both Yao et al. (2012) and our pattern of glacier

changes in the inner Himalayan ranges (see also Sakai and Fujita, 2017). In particular the stable glacier elevations in our unit HK1 — between areas of glacier loss in the Hindu Kush and the particularly negative western Himalaya (units H4–H6) — are backed up by their modelled stable ELAs. According to MERRA-2 data (but not ERA Interim), the area experienced an increase in summer precipitation between the 90s and 00s (Figs 8a, 8b). The particularly negative surface elevation change in the western Himalaya has previously been attributed to rapidly shrinking accumulation areas, seen in rising firn lines in

Landsat images (Kääb et al., 2015, area called Spiti Lahaul). Kääb et al. (2015) see the same pattern for the strongly negative glacier evolution in Nyainqêntanglha Shan/Hengduan, which has low-lying accumulation areas. Thus, once the accumulation area becomes too small or disappears entirely, also abundant or increasing precipitation cannot compensate for melt due to increased temperatures (Sakai and Fujita, 2017).

### 5.6  Dissimilar glacier behaviour in the Karakoram/Kunlun Shan

The zonation we present here is the result of a compromise between within-unit glacier similarity, representative sampling, and stable glacier surface change rates. In the Karakoram/Kunlun Shan area, this approach is clearly more appropriate than sample grouping into a regular grid. The latter results in large uncertainties in the glacier elevation change signal (Fig. 2b), since grid cells include both the thinning signal south of the central Karakoram and thickening signal in the Kunlun Shan.

In the Karakoram, we see indications of both surging glaciers and glaciers recovering from a surge. In most units, such as

K1–K3, the surface elevation change signal is different in the upper 50% elevations compared to the ablation areas. This is in line with e.g. Gardelle et al. (2012a, 2013), who find that most of the glaciers in this area were in some stage of a surging cycle in the ICESat decade. Our units are just large enough not to be dominated entirely by a retreating or rapidly growing tongue of one single large glacier, but rather provide an average of these locally different signals. After ensuring correct hypsometry sampling, the surface elevation changes of the different units in the area agree well. We find evidence of surging glaciers also

in other areas, such as the Zhongfeng glacier in the Western Kunlun Shan (unit KS1) (Ke et al., 2015). ICESat does not sample the tongue of Zhongfeng glacier (whose surface might be rising) and the negative elevation changes dominate the signal in the unit — which does not fit the otherwise positive elevation change of the surrounding units. Aggregated in larger spatial units such as a regular grid, this local peculiarity is not visible. Whether such signal is representative for all glaciers in a unit or not would require complete geodetic analysis of all glaciers and also a longer time span.

### 30  5.7  Varied pattern in Tien Shan

Glacier evolution in the Tien Shan has shown a spatially diverse pattern already in the last decades of the 20th century (Narama et al., 2010; Farinotti et al., 2015). Together with contributions from northerly areas, the Westerlies are the source of precipitation for the entire region (Bothe et al., 2012), but there are different climatic sub-regions: glaciers in the Western Tien Shan (and

Pamir Alai) receive precipitation mainly in winter, the northern and northeastern ranges both in winter and summer, whereas the inner ranges are of the spring/summer accumulation type (Sorg et al., 2012). In the (north)western Tien Shan, our zonation does not consider this transition from winter-only to summer/winter precipitation due to too low sample numbers for a finer zonation in this area.

Narama et al. (2010) suggest that glaciers of the outer ranges — which receive more precipitation — are melting faster since they have a higher mass turnover and their tongues are at lower elevations. They see such a pattern in 2000–2007 glacier shrinkage which was more pronounced in the Western/northern Tien Shan than in interior areas such as the southeastern Fergana Range or At-Bashy Range at the transition to the Pamir. Our thinning rates do not confirm this — precisely in this latter area (unit P3), we find the most negative glacier surface elevation changes in the entire region (converted to mass change:

$-1.04 \pm 0.23$ m w.e.). The modelling study of Farinotti et al. (2015) suggests spatially highly varying glacier reactions in the last few decades in that area (their coarser zonation in the Central Tien Shan does not allow direct numerical comparison with our results).

ICESat suggests moderate thinning for the north-eastern Borohoro range, in particular the central part at higher elevations (TS1, converted: $-0.09 \pm 0.18$ m w.e. a$^{-1}$, upper 50% glacier elevations thickening in Fig. 3a). Farinotti et al. (2015) found

that the central parts of the range receive 50% more summer precipitation compared to the rest of the range, and modelled $-0.17 \pm 0.24$ m w.e. a$^{-1}$ for 2003–2009 for a slightly larger area than our most central unit.

In the inner Tien Shan, our elevation change rates vary on a small spatial scale. Reconstructed annual mass balances (Kenzhebaev et al., 2017; Kronenberg et al., 2016) and DEM differencing/modelling studies in the area (Fujita and Nuimura, 2011; Shangguan et al., 2015; Barandun et al., 2018) match the range of our thinning signal. Our zonation does not consider glacier

aspects which seem to play an important role in explaining glacier melt over this region (Farinotti et al., 2015). For the glaciers in the Aksu-Tarim catchment in central Tien Shan, Pieczonka et al. (2013) found a decelerated mass loss between 1999 and 2009 ($-0.23 \pm 0.19$ m w.e. a$^{-1}$) compared to earlier decades, which supports our only slight thinning on the northern slopes of the Tarim basin. Our units with less thinning resemble the pattern of glaciers with little long-term changes by Farinotti et al. (2015, modelled) — except for our slight thickening signal in the southern Halik Shan on the northeastern edge of the

Tarim basin. The few glaciers in this unit are small and lie at lower elevations which would make them prone to fast melting in a warming climate. A possible explanation is a false or exaggerated trend due to snow cover in late 2008, as correcting the December 2008 campaign accordingly effectively removes our thickening signal ($0.02 \pm 0.31$ m w.e. a$^{-1}$, Supplementary Fig. S1c).

## 6 Conclusions

We present a complete and consistent estimate of glacier surface elevation changes for entire High Mountain Asia (HMA) based on ICESat data for 2003–2008 and relate the spatial pattern to lake volume and precipitation changes on the Tibetan Plateau (TP). For the ICESat analysis, our new spatial zoning better reflects different glacier setting in particular in relation to orographic effects, and updated methods ensure that biases present in earlier ICESat studies are removed. The study addresses

several new aspects of the spatial pattern of glacier changes and stresses in particular the role of precipitation and elevation sensitivity of glaciers in different parts of HMA. To confirm underlying precipitation changes on the TP with an independent approach, we estimate the 1990–2015 change in total water volume from all endorheic lakes on the TP, based on variations in both areal extent and water surface levels. The latter work results in volume change time series of >1300 lakes, much more than available so far. In more detail, we conclude:

– Only carefully delineated spatial units show local patterns of glacier change that are diluted or hidden if samples are gridded. On a larger scale, the pattern we find in this study agrees with previous regional estimates based on ICESat — but provides finer detail. The new zonation and improved bias control in this work stretches the applicability and precision of ICESat-derived elevation changes in rough and glacierised terrain further than was the case for previous studies.

– The pattern of glacier changes is spatially varied because of differences in the glaciers' elevations and sensitivity to climate changes (Sakai and Fujita, 2017; Kapnick et al., 2014). Together with glacier elevations, precipitation distribution and changes are able to explain large parts of the spatial variability of the glacier change pattern observed for 2003–2008.

– An almost stepwise precipitation increase on the TP, Kunlun Shan and possibly also the Tarim Basin between 1995 and 2000 is clearly visible from MERRA-2 reanalysis data and coincides in time with observed changes in lake water volume. The precipitation increase is able to fully explain 2003–2008 glacier thickening in an area centred over the Kunlun Shan. The boundary between positive and negative glacier changes is rather sharp in the Kunlun Shan and lies north of the main Karakoram range. It is more gradual on the TP, and glaciers on the northern slopes of the Tarim Basin were close to balance.

– Lake volume changes on the TP reflect a clear and comparably sudden increase of water availability from ca. 1997 through ∼2010 for the northern and eastern TP, but only minor changes in the southwestern TP and Qilian Shan. The observed lake changes correspond to a precipitation equivalent of 6–7 mm a$^{-1}$ for the northern TP and 25 mm a$^{-1}$ for the eastern TP, from decadal averages between the 1990s and 2000s. MERRA-2 reanalysis data suggests the change is exclusively be driven by increased summer precipitation of 34–100 mm decadal difference between the 1990s and 2000s. ERA Interim reanalysis data suggests a smaller precipitation increase for a smaller spatial area that does not explain lake growth and glacier thickening equally well.

– The magnitude of lake volume change, glacier mass balance and precipitation changes agree with each other when accounting for evaporation. Increased influx from glacier mass loss may in some areas have contributed to lake growth but cannot explain it, as the zone of lake growths roughly coincides with the zone of positive glacier mass balances or dynamical glacier geometry change.

– Glaciers on the TP changed their geometry during 2003–2008. In the northeastern TP/western Kunlun Shan, upper glacier surface elevations were stable while tongues were growing. Further south/east, upper elevations were thickening

while the tongues were thinning due to both increased accumulation and melt. The further southeast on the TP, the stronger the glacier thinning rates. Glaciers in the Qilian Shan were only moderately losing mass.

- Along the entire Himalayan Range, glaciers on the first orographic ridge were thinning less than those further back in a drier climate, likely due to abundant precipitation on the first ridge, which causes equilibrium line altitudes (ELAs) to be at lower elevations. Precipitation and ELA gradients might be very steep in the outermost ridges of the Himalaya.

While the glacier change pattern presented in this study is robust and well explained by glacier sensitivities to climate change, our unit boundaries might not match areas of consistent glacier changes everywhere, despite best efforts. Low ICESat sample density prohibits a further refinement in areas with sparse glacier coverage. Other remote sensing data with finer spatial resolution could improve the pattern — for example DEM differencing from ASTER stereo-imagery (Brun et al., 2017) and other spatially extensive data available for the last decades, or also ICESat-2, once this data becomes available. Combinations of remote sensing products for precipitation, snow and atmospheric parameters as well as improved reanalysis data could help to determine precipitation numbers with more certainty in Asia's water tower.

*Code and data availability.* ICESat data are freely available from NSIDC and NASA, the SRTM DEM and Landsat data from USGS, the MERRA-2 reanalysis data from NASA Goddard Earth Sciences Data and Information Services Center, ERA Interim from the European Centre for Medium-Range Weather Forecasts, the Global Surface Water dataset within Google Earth Engine. The derived ICESat zonation is available as a data supplement to this publication.

**Appendix A: Data**

**A1    ICESat elevation data**

The NASA Ice, Cloud and land Elevation Satellite (ICESat) measured the Earth's surface elevations in two to three campaigns per year from 2003 to 2009. The campaigns were flown in northern autumn (∼October–November), winter (∼March), and early summer (∼June). Autumn is overall the driest season in HMA, and ICESat's autumn elevation samples on glaciers thus fall to a large extent on ice and firn rather than fresh snow. By contrast, snow falls in March/June in parts of HMA. ICESat's Geoscience Laser Altimeter System (GLAS) sampled surface elevations within ground footprints of ∼70 m in diameter (Schutz et al., 2005). Elevation samples are separated by ∼170 m along ground tracks/orbits but up to 75 km between orbit paths in HMA. The ground track pattern was not repeated exactly during each overpass, as the near-repeat orbit mode was not activated at lower latitudes (Schutz et al., 2005). Rather, individual ground tracks lie as far as 2–3 km from the reference ground track in HMA. A direct comparison between ICESat elevations is thus difficult in the region. Instead, double-differencing techniques are applied, i.e. comparing ICESat elevations with a reference DEM to receive elevation differences and analysing their subsequent evolution over time (Kääb et al., 2012; Gardner et al., 2013; Neckel et al., 2014; Kääb et al., 2015; Ke et al., 2015).

Here, we use GLAS/ICESat L2 Global Land Surface Altimetry HDF5 data (GLAH14, release 34) which is optimised for land surfaces (Zwally et al., 2012). From comparison with reference DEMs, elevation uncertainty of GLAH14 data was found to be on the order of decimetres to metres in mountainous terrain in Norway (Treichler and Kääb, 2016). Elevation biases and inconsistencies throughout ICESat's lifetime are of centimetre to decimetre magnitude and thus negligible compared to uncertainties from the underlying terrain and biases in the reference DEM (Kääb et al., 2012; Treichler and Kääb, 2016).

## A2  SRTM DEM

The DEM from the Shuttle Radar Topography Mission (SRTM, Farr et al., 2007; Farr and Kobrick, 2000) is a consistent DEM in the HMA region. We used the C-band, non-void-filled SRTM DEM version at 3 arc-seconds resolution (SRTM3, corresponding to 92 m in y, and 66–82 m in x-direction at 45/28° N) which is accessible from the U.S. Geological Survey at https://dds.cr.usgs.gov/srtm. The SRTM DEM used here is a product of single-pass C-band SAR interferometry from images acquired on 11–22 February 2000 (Farr and Kobrick, 2000). SRTM DEM nominal vertical accuracy is of the order of metres (Rodriguez et al., 2006). Treichler and Kääb (2016) found spatially varying vertical offsets on the order of metres to decimetres in mountainous terrain in Norway. They attributed the vertical biases to the fact that the SRTM DEM is a composite from several individual images and overpasses, and likely processed in (unknown) spatial sub-units. Offsets caused by shifts of sub-units were not removed by global DEM co-registration, but the bias/uncertainties caused by them are within the nominally stated accuracy. On glaciers, larger elevation uncertainties are to be expected due to penetration of the C-band signal into ice and, even more so, into snow. Also dry sedimentary soils may be subject to radar penetration. The penetration is estimated to be in the range of several metres for glaciers in HMA (Gardelle et al., 2012a; Kääb et al., 2012, 2015).

The vertical offsets from DEM shifts or penetration increase the uncertainty of surface elevation changes — possibly also for ICESat-based studies, if the spatial pattern of SRTM DEM offsets interferes with ICESat's spatial sampling pattern (Treichler and Kääb, 2016, 2017). As an alternative elevation reference, we used the SRTM DEM at 1-arc-second resolution (SRTM1) from https://earthexplorer.usgs.gov. The 1-arc-second DEM has undergone fewer revisions than the 3-arc-second DEM, making the data not necessarily superior, and most data voids are filled in with other elevation data that have different time stamps. We therefore excluded the data void areas contained in the 3-arc-second DEM version also in the SRTM1 DEM to ensure that we only use original elevation data from February 2000.

Further, we did not explore or use the recently published TanDEM-X global DEM as it was not available during our processing. It remains to be investigated how potential advantages of this DEM (larger coverage, less penetration than C-band) balance potential disadvantages (longer time difference to ICESat period, temporal inconsistency from stacking). Also due to temporal inconsistency and substantial voids, we did not use the ALOS PRISM World DEM (AW3D) or the WorldView satellite optical stereo HMA DEM.

## A3  Precipitation data

As an estimate for regional and temporal precipitation patterns for the years 1980–2015 we use data from the Modern-Era Retrospective analysis for Research and Applications, version 2 (MERRA-2 Gelaro et al., 2017) at resolution of 0.625° x 0.5°

in lat/lon and available at https://disc.sci.gsfc.nasa.gov/mdisc from the NASA Goddard Earth Sciences Data and Information Services Center. We also use the ERA Interim reanalysis (Dee et al., 2011) at T255 spectral resolution (0.7° lat/lon), available from the European Centre for Medium-Range Weather Forecasts at http://apps.ecmwf.int/datasets/. We use monthly summarised values of the variables total precipitation (PRECTOT / tp), snowfall (PRECSNO / sf) and evaporation (EVAP / e) from MERRA-2's surface flux diagnostics dataset tavg1_2d_flx_Nx (GMAO, 2016) and ERA Interim's Monthly Means of Daily Forecast Accumulations, respectively. Due to the scarcity of observations in HMA, reanalysis products are less constraint and have higher uncertainties in our study area than in more densely populated areas of the Earth. The two chosen reanalysis products have been found to model precipitation and snowfall comparatively well (Reichle et al., 2017a, b). The High Asia Reanalysis (HAR, Maussion et al., 2014), a product optimised for the TP region and with much finer spatial resolution, is unfortunately only available for the time period 2001–2011 which is too short for our study with respect to the lake volume changes investigated.

The meteorological stations included in this study were chosen because they are closest to the area with reported glacier mass gain. We are not aware of any meteorological measurements on the northwestern TP.

### A4  Global Surface Water Dataset

The Global Surface Water dataset (Pekel et al., 2016) is a classification of the entire Landsat archive into monthly and annual maps of surface water (https://global-surface-water.appspot.com). The data is available within Google Earth Engine (Gorelick et al., 2017). To map the changing extents of Tibetan lakes, we used the variable occurrence which provides the classes no data, no water, water (for both monthly/annual data), and seasonal water (for annual maps only). Pre-2000 coverage is poor for years with little Landsat data, for our areas of interest: 20–75% no-data pixels in 1990, 1991, 1995, 1997 and 1998 (Pekel et al., 2016).

### Appendix B:  Methods for glacier volume change

We follow the double-differencing method explained in Kääb et al. (2012) and Treichler and Kääb (2016). ICESat data and individual SRTM DEM tiles were converted into the same geographical reference system, co-registered (Nuth and Kääb, 2011), and reference elevations for ICESat footprint centres retrieved by bilinear interpolation. The difference between ICESat and SRTM elevations is further referred to as dh. Double differencing, i.e. fitting a linear trend through dh from several years, reveals how much the surface elevation has changed on average over the time period studied.

ICESat samples were reduced to those within a 20 km buffer around RGI glacier outlines. To avoid inclusion of off-glacier elevation samples in our glacier surface change analyses (see introduction), we classified all ICESat footprints manually into *glacier* and *off-glacier* samples, using the most snow-free Landsat images from ca. 2000–2013. Samples on water and clouds ($|dh| > 100$ m) were excluded. Samples on glacier borders were also excluded, to avoid inclusion of 70 m footprints that only partially fall on ice and because glacier areas could have changed in the course of 2003–2008 (Treichler and Kääb, 2016). To

compute statistics per glacier, we also classified the samples based on glacier outlines of the newest version of the RGI (version 6, RGI Consortium, 2017).

To test the sensitivity of biased dh at either end of the studied time period, we do not only compute a robust linear regression, which is commonly used for ICESat glacier applications (Kääb et al., 2012), but also a t-fit (Treichler and Kääb, 2016) and a
non-parametric Theil-Sen linear regression (Theil, 1950; Sen, 1968). Both alternative robust fitting algorithms better fit our dh distribution and are commonly used for datasets with large natural variability and measurement errors.

We find little difference between robust and t-fits, and slightly larger (but no systematic) differences when using Theil-Sen linear regression. The trend slopes from the three methods agree on average within 0.1 m/a and differences always lie well within trend error estimates. Our final estimate per spatial unit thus corresponds to the average of the three trend methods.

**B1   Zonation**

As seen in Kääb et al. (2015), grouping of ICESat samples into a regular grid without a-priori knowledge results in a blurring of local glacier change signals. Since such local signals consist of a specific dh magnitude and evolution over time which should be governed by climatic or topographic drivers, we tried to derive a more realistic spatial division from the ICESat samples directly, using glacier statistics, dh and iterative clustering. This approach was not successful: the number of (semi-
quantitative) statistical parameters turned out to be too large and dh vary too much spatially, not least due to bias. We thus carefully delineated spatial units manually. Zones were drawn by hand to avoid splitting any glacier between several zones. In particular, we paid special attention to orographic barriers. Rather than roundish zones across the entire Himalayan range, we chose elongated zones around mountain ridges. Size, length and width of spatial units (i.e. how many parallel ridges) were largely determined by ICESat sample numbers and the condition of representativeness. For example, we included both the
windward and leeward side of a Himalayan range as there are very few glacier facing south (i.e. windward), and we suspected that leeward accumulation areas close to a mountain peak might still receive more precipitation from turbulences than the dry, leeward valley bottoms (Immerzeel et al., 2014). We are very aware that our zonation is a subjective one and open to discussion. In some parts, other operators will likely come up with modified zones. However, our zonation is based on carefully-applied expert knowledge, and we are convinced it displays the 2003–2008 HMA glacier elevation changes with a spatial resolution
and precision that reflects the optimum that is feasible from ICESat over such a mountainous and heterogeneous region.

**B2   Glacier hypsometry**

We compute the relationship between glacier dh and elevation (hereafter called dh–elevation gradient) by fitting a robust linear regression through individual glacier samples' dh vs. elevation. Greater radar penetration in the accumulation areas and more prominent melting of tongues steepen dh–elevation gradients (e.g. Vijay and Braun, 2016; Ragettli et al., 2016). Representative
elevation sampling through time and in relation to local glacier hypsometry is thus very important. Our primary approach to improve sampling hypsometry is to enlarge spatial units, but in some areas this would have led to considerably reduced glacier similarity within the unit. To account for these conflicting cases, we computed four different corrections and compared the such-adjusted results: (A) correcting the slope of the glacier elevation-change trend for the effect of a positive/negative elevation

trend in time, i.e. correcting for the case where ICESat consistently samples higher/lower elevations (smaller/larger dh) with time (Kääb et al., 2012, suppl.); (B) correcting individual dh for the effect of elevation, i.e. computing the expected dh from the dh–elevation gradient and the individual elevations sampled, and removing the expected dh values from the measured dh values; (C) filtering of the samples of each ICESat campaign to match the hypsometry of the glaciers within each spatial

analysis unit; (D) assigning weights to samples depending on their elevation so that they match the glacier hypsometry, i.e. analogue to C but without removing any samples.

All four corrections are here applied to all units, and both for glacier and off-glacier samples separately. Methods A and B are based on the method used in Kääb et al. (2012, 2015). If ICESat consistently samples lower (or higher) elevations than the reference hypsometry, methods A and B will not correct for this — they only correct elevation-induced bias relative to

the mean sampled elevations of all campaigns. Methods C and D, however, adjust the hypsometry so that it should become representative for the glacier elevations in the unit. For 18 units, the difference in derived surface elevation change between the 'standard method' (average of all methods A-D) and only applying the latter methods (average of methods C and D) exceeds $0.05\,\mathrm{m\,a^{-1}}$, and at the same time, average glacier elevations sampled in these units is also $> 50$ higher or lower than average glacier elevations for this unit (SRTM elevations within RGI glacier outlines). For these units with systematic elevation

missampling, we used the average of methods C and D only. To 5 of the affected units we also applied the cG correction (see below).

## B3   Correction of vertical bias

To remove local systematic elevation bias, we compute a per-glacier elevation correction cG corresponding to the median dh for each glacier (i.e. subtracting the median dh for each glacier from each corresponding dh). In the study of Treichler and

Kääb (2016), the correction successfully reconciled annual ICESat-based glacier elevation changes with mass balance time series from in-situ measurements. Also in the present study, cG-corrected dh (in combination with above hypsometry methods A–D) remove the effect of a varying spatial composition of elevation offsets. However, the correction results in lower sample numbers and removes parts of the signal where some glaciers are only sampled in the beginning and some other glaciers only in the end of the ICESat acquisition period. There, the correction shows a tendency to erroneously flatten out linear trends. We

thus apply cG only where the opposite is the case and trends become considerably ($> 0.05\,\mathrm{m\,a^{-1}}$) steeper after cG correction. This is the case for 21 units. To limit the effect of potential bias from lower sample numbers, our final trend estimate for these units is the average of the 'standard method' with and without application of cG, respectively. The final thinning/thickening rates of the affected units differ from the 'standard method' by on average $0.08\,\mathrm{m\,a^{-1}}$ and range from $-0.37$ (unit HS) to $+0.15\,\mathrm{m\,a^{-1}}$ (a unit in the central Karakoram range).

The onset of winter snowfall might cause erroneously positive dh in the December part of the split autumn 2008 campaign for parts of HMA: areas under influence by the Westerlies (Tien Shan, Pamir, Karakoram, western Himalaya) or winter precipitation in Nyainqêntanglha Shan/Hengduan Shan (Maussion et al., 2014). We estimate the influence of this according to the method of Treichler and Kääb (2016).

## Appendix C: Methods for lake volume change

We compute annual water volume change of the Tibetan lakes by multiplying annual lake areas with water level changes from repeat water surface elevations for each year over the period 1990–2015. Maximum annual lake extents are obtained directly from the Global Surface Water data set by exporting bitmaps of annual water occurrence over the entire TP, using the web API of Google Earth Engine. The data is exported at a resolution of $50\,m \times 38$–$44\,m$ in lat/lon (corresponding to 0.00045 degrees). Subsequently, we retrieve the corresponding lake surface elevations in two ways: a) from SRTM DEM elevations of the lake shore by computing the median of interpolated DEM elevations for lake shore cells for each areal extent, and b) directly from ICESat footprint elevations on the lake areas for those lakes where ICESat data is available. To extend the lake elevation time series from method b) beyond the ICESat period of 2003–2009, we compute the area–surface-elevation relationship for each lake by robust linear regression and apply this function to the areal extends of the years before and after the ICESat period. We extract the relationship both for annual time series and individual ICESat campaigns (2–3 campaigns each year, using the monthly water classifications). The so-extrapolated surface elevation values generate complete 1990–2015 time series for both areal extent and lake levels from SRTM and ICESat data, respectively. Our method is in parts similar to the methods used by previous studies investigating lake volume changes on the TP from satellite data (e.g., Zhang et al., 2011; Kropáček et al., 2012; Song et al., 2013; Zhang et al., 2013; Song et al., 2015; Zhang et al., 2017) but the inclusion of a DEM for deriving shoreline elevations, and thus lake water levels, in addition to altimetry data enabled us to produce volume change time series for one order of magnitude more lakes than derived previously.

We apply our procedure to the 1364 endorheic lakes on the TP and in the Qaidam Basin (Fig. 1) with a maximum lake extent of $> 1\,km^2$. We generated here our own lake database since we found that existing collections, such as the Global Lakes and Wetlands Database (Lehner and Döll, 2004), are lacking numerous lakes that likely only emerged during the last two decades. Consulting satellite imagery like Landsat data, we manually adjusted our lake database to remove delta-like seasonal wetlands from water inflow on sloping terrain from the lake masks, we excluded non-endorheic lakes (visible outflow), and we excluded inundated areas affected by human interventions, e.g. for salt production (in total 133 wetlands, not included in the above number). For spatial aggregations, computation of relative numbers per lake and for plotting, we use the median lake areas from the 1990–2015 annual lake extents.

### C1  Uncertainties and filtering

Uncertainties associated with the lake data used include misclassification of water area in the Global Surface Water dataset (Pekel et al., 2016), lake surface elevation errors and local bias in the SRTM DEM, and bias in ICESat surface elevation measurements. For each lake and year, we compute the percentage of missing data (e.g. from cloud cover or classification voids), and years with $< 95\%$ of data coverage within the lake masks are excluded from further analyses. Lake time series that, after removing these years of insufficient coverage, do not contain any data from the 1990s are excluded entirely. For ICESat-derived lake levels, only lakes with measurements from at least three laser footprints each from at least five years are considered. Data from the 90s have higher uncertainties in extracted/extrapolated lake levels due to a) the implicitly assumed

bathymetric profile using area–lake level scaling for years without ICESat data; and b) because the SRTM DEM was acquired in February 2000: While lake areas vary seasonally and we use annual maximum areas, the effect of extracting SRTM lake elevations for lake areas smaller than during SRTM data acquisition is that some pre-2000 SRTM lake levels may be too high, resulting in too small dV. Despite the lake areas and surroundings being extremely flat, SRTM DEM cells indicate up to 10 m elevation differences between neighbouring cells in a seemingly random way, and the SRTM DEM turns out to be the dataset within our lake change analysis with the greatest uncertainties. Potential explanations for the DEM elevation uncertainties are penetration of C-band radar into sandy ground and unknown processing steps during DEM production to mask/interpolate water-covered areas without radar backscatter. For some lakes, SRTM DEM errors result even in negative area–lake-surface elevation relationships, i.e. lake shore elevations seemingly decrease for expanding lake areas which is physically not plausible. We therefore excluded all lakes with either a negative area–lake-elevation relationship or where the 26-year linear trends for lake area and lake surface level do not have the same sign. This is done both for ICESat- and SRTM-derived lake level estimates. The overall error for a decadal average lake volume stage is estimated as the standard error of the mean, and for decadal differences propagated as the root of the sum of squares of the two errors (RSS).

## C2 Endorheic basins

We summarise and spatially distribute the water volume changes based on endorheic catchments of the USGS HydroSHEDS dataset at 15 arcsec spatial resolution (Lehner and Döll, 2004, https://hydrosheds.cr.usgs.gov) . However, many catchments only contain a single lake and exact catchment areas are not well defined on the TP (e.g., in very flat areas, Lehner et al., 2008), and the spatial resolution of the HydroSHEDS dataset is in parts too coarse to correctly attribute the lakes of our lake dataset to the correct catchment. Therefore, we manually controlled and adjusted the endorheic catchment borders using the finer topography of the SRTM DEM at 3 arcsec resolution as well as Landsat imagery to detect surface water exchange between lakes/catchments, and aggregated the catchments to larger basins of comparable size, consisting of in average 5 catchments.

We define the total lake area per catchment (and basins) as the sum of the 1990–2015 median lake area of all lakes within the spatial unit, also including the endorheic inundated areas confined by human infrastructure mentioned above, which are otherwise excluded from analyses. To compute total water volume change per catchment, we assume that lakes excluded from the analysis (see previous subsections) behaved the same way as the average of the lakes we have sufficient data for, and subsequently scale the total volume change accordingly. For total water volume change from decadal averages, we compute the error as the sum of the errors of all individual lakes' volume change (see above), again scaled according to the share of total lake area we have sufficient data for. This conservative approach of adding errors (instead of root-sum-of-squares, for instance) includes as a worst case the full correlation of the behaviour of all contributing lakes.

## Appendix D: Discussion of biasing influences on ICESat glacier surface elevation change

Representativeness of samples within spatial units is the key requirement for robust glacier thickening/thinning estimates. However, we found that enlarging spatial units was not always the best remedy to ensure sample representativeness: In some

areas this would have considerably reduced glacier similarity within the unit. Applying a regular grid can have the same effect. Consequently, only carefully adapted zones can show local peculiarities that are otherwise diluted.

Especially for small units with few samples, careful consideration of how potentially biasing factors interplay is important. Our use of four different methods to ensure correct hypsometry sampling makes the results very robust. The overall pattern is not affected by zonation, small changes in sample composition (RGI outlines), or reference DEM (here: SRTM1). Of all corrections, the most essential requirement is therefore that the regional glacier hypsometry is sampled appropriately, also over time. Locally, however, the different methods and corrections can result in considerable differences between glacier thickening/thinning rates. Especially where ICESat data is used on a local scale or as input for modelling studies, we strongly recommend to carefully assess the difference between hypsometry corrections, the effect of our per-glacier correction cG, and the influence of snow cover, in order to ensure a representative estimate and appropriate uncertainty.

Our snow correction affects trends significantly. In southern Norway, the study region for which the correction was developed, it removed a positive off-glacier trend but did not affect the glacier trend (Treichler and Kääb, 2016). Our results in HMA show that trend fitting methods are surprisingly sensitive to a lowering of the last (half) campaign, no matter which trend fitting algorithm is used, and for both off-glacier and glacier dh. In contrast, if the same correction is applied to a campaign between 2004 and 2007, trends only change marginally. The exercise shows that November/December 2008 snow fall has the potential to erroneously decrease ICESat-derived glacier thinning rates, in particular in Tien Shan, Pamir, Hindu Kush, Nyainqêntanglha Shan/Hengduan Shan and maybe also the outer Himalayan ridges (Supplementary Fig. S1). We therefore recommend to assess the bias potential of December 2008/October snow fall for ICESat studies on a smaller spatial scale. Also, we advise not to rely on ICESat's March campaigns for glacier studies wherever snow is falling in winter in the northern hemisphere.

ICESat elevations have previously been used to estimate SRTM penetration (Kääb et al., 2012, 2015; Shangguan et al., 2015). On glaciers where no ICESat data is available, dh–elevation gradients of larger spatial units — such as in this study — could improve the estimated elevation dependency of penetration.

*Author contributions.* D. Treichler jointly designed the study with A. Kääb, performed all data analyses and prepared the manuscript. A. Kääb contributed to data interpretation and edited the manuscript. N. Salzmann and Ch.-Y. Chu contributed to analyses and interpretation of the climate/precipitation signals and lake water volumes, and the joint interpretation of the data.

*Competing interests.* The authors declare no competing interests.

*Acknowledgements.* The study was funded by the European Research Council under the European Union's Seventh Framework Programme (FP/2007-2013)/ERC grant agreement no. 320816, the ESA project Glaciers_cci (4000109873/14/I-NB) and the Department of Geosciences, University of Oslo. We are very grateful to NASA and USGS, ECMWF, and the China Meteorological Administration for free provision of the ICESat, Landsat, MERRA-2 data and the SRTM DEM version we used, ERA Interim data, and meteorological station data, respectively.

Special thanks go to Patrick Wagnon, Joe Shea and the Cryosphere group at ICIMOD, Nepal, and Martin Hölzle, Martina Barandun and the Physical Geography group at the University of Fribourg, Switzerland, for their valuable input on the spatial zonation. We thank the two anonymous reviewers whose comments and suggestions helped to improve and clarify this manuscript.

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
