# Peer review of "Recent glacier and lake changes in High Mountain Asia and their relation to precipitation changes"

_The Cryosphere, 2018_

## Referee Comment (RC1) · Anonymous Referee #1 · 24 Jan 2019

General

The authors present an interesting study and they analyze surface elevation changes in High Mountain Asia with ICESat data between 2003 and 2008. They hypothesize that the positive glacier mass balances found in the eastern Pamir, Kunlun Shan and the central TP can be explained by a step-wise increase in precipitation. They approximate the precipitation change by quantifying changes in lake volume of endorheic lakes, station and reanalysis data. I believe the study definitely has scope to be published in the cryosphere, but I find that the conclusions drawn are too strong and are not supported well enough by what the (uncertain) data shows. I have identified the following issues

that need to be addressed before the paper is acceptable for publication:

1. Previous work has aggregated surface elevation changes on a 1 degree or 2 degree grid. In the present study the authors have made their own delineation, which they acknowledge to be subjective. The procedure for delineating the spatial units is not clearly described (p6, l17-24). It comes across as if polygons are drawn around region where trends are most clear and obviously the resulting zonal map (Fig. 2A) looks better than the gridded map (Fig. 2B). The use of the zones needs better justification and they have to be objectively defined ideally without prior knowledge about the ICESat trends.

2. The lake changes are solely attributed to precipitation changes and I have some doubts about this assumption. I think a potential important factor can be the change in permafrost. Much water is stored in frozen form in the soil. An increase in the active layer as a result of rising temperatures may also considerably impact the lake water balance. However this is not at all discussed, and temperature trends are not mentioned either. Therefore I recommend to include references to changes in permafrost hydrology and to quantify spatially also the temperature trends based on the reanalysis datasets. Furthermore the assumption that most lakes are endorheic is quite strong. The subjective description of how the HydroShed dataset has been manually modified is a bit worrying (p7, l31-34). It would be recommendable to clarify this.

3. MERRA-2 is a reanalysis dataset which is known not to perform very well in high mountain Asia, yet strong conclusions are drawn based on the projected changes in precipitation. The results of ERA-Interim are largely ignored since they not match as well to the observed lake and glacier changes. It is recommended to better justify the use of MERRA-2 and show a comparison with the station data or provide another argumentation why this reanalysis dataset should be used. It may also be worthwhile to use the recently released ERA5 dataset which is the high resolution successor of ERA-Interim.

4. The use of actual evapotranspiration from MERRA-2 to derive a lake basin water balance is questionable and highly uncertain. The uncertainty needs to be discussed and quantified or ideally an ensemble of re-analysis products should be used. The authors may even consider leaving out the whole reanalysis part given its uncertainty. Linking the lake and glacier dynamics is already exciting enough.

5. The authors conclude that the lake changes match the glacier surface elevation change very well. I am not sure if I agree. Table 1 shows positive lake volume changes everywhere, while the glacier mass balance shows contrasting trends across the region. In addition the periods do not match (1990-2000 versus 2003-2008). The same holds for Figure 4. I do not see many similar patterns between Figure 2A and Figure 4. The lake growth very clearly starts in 1995 (Fig 5), but the increase in precipitation occurs about 5 years later (Fig. 6 and 8), so that does not make sense to me.

6. Glacier changes are explained only from an accumulation (and precipitation change) perspective. However a glacier mass balance is the results of accumulation and ablation. Total precipitation may increase, but if the temperature increases as well that may still result in less snow. In addition an increase in temperature may also enhance the melt and other energy balance terms may change. Recent work explains the Karakoram anomaly as a result of more summer snow fall and less melt due to less incoming shortwave radiation due to more clouds and a higher albedo(de Kok et al., 2018). Another important study identified the Karakoram Vortex, which draws cold air into specific part of the region (Forsythe et al., 2017). None of these factors are considered.

Specific comments

P2, l25: I recommend a more detailed comparison with the Brun et al., (2017) results

P3, l6-7: HMA does not have a typical winter snow fall – summer melt cycle. While this may be true in the west, in the monsoon dominated areas the winters are generally dry and there is synchronous ablation and accumulation. Therefore (high altitude) summer snow events may also cause a bias. In addition I wonder given the type of

trend analysis used, why a single anomalous event in autumn 2018 causes such a bias.

P4, l5: What is meant by precipitation availability? Just precipitation is enough I would suggest.

P5, l13: insert here a paragraph on ablation and radiation regimes across HMA?

P5, l20-25: not sure if it adds value to mention what has not been used. It is absolutely fine to use SRTM.

P5, l29: some validation of MERRA5 is required. Large cold biases in reanalysis datasets are very common and this may have very large effects on the modelled snow-fall for example.

P5, l4: add reference to the Global Surface Water dataset

P6, l14-16: why use three different methods and then use the average? This assumes each method performs equally well. Are there no arguments why a certain method is preferred in this case?

P6, l16: Same. Why use four different ways of hypsometry corrections and then take the average?

P7, l4: Snow does not fall only in winter in HMA, so how are other campaigns influenced?

P7, l17-20: Again why multiple methods?

P7, l20-24: If most lakes are growing and the reference DEM is SRTM (∼2001) or IceSat (2003-2009) then how is the water volume change reconstructed prior to this period as there is no information about the lake bed elevation below the water. A discussion regarding the uncertainty of using an "above the water" volume-area scaling would be useful.

P8, l5-10: the authors indicate that the reanalysis data is not accurate, but still it is used to draw strong conclusions.

P12, l7-8: Why is the ICESAT based lake level change 1.55 times as large? Does that point towards a systematic difference between SRTM and ICESat in off-glacier areas?

P12, l8: lake growth = water level increase?

P13: Fig. 4: What is meant by median lake area? Express 4b as mm/year to make it comparable to precipitation rates?

P14, Figure 5: Very interesting to see the abrupt increase from 1995 except for the Qilian Shan region.

P15: Instead of the data in table 1 I suggest to sync the periods and show the 2003-2008 glacier mass balance, lake volume change, re-analysis precipitation change and re-analysis precipitation minus evapotranspiration change.

P16. Fig6: Add the reanalysis data for the same pixel as the stations to assess its validity? Stepwise increase (if significant) occurs around 2000 which is 5 years later than the lake increase. Same for Fig. 8. One solution could be to look at trends and test their significance rather than focusing on the "step-wise" increase.

P19 l30-31: very thin basis for this conclusion.

P22, paragraph 5.3: very interesting finding that the southern slopes have less negative mass balances. It seems to be related to a higher mass turnover and a reduced sensitivity of the mass balance to temperature changes.

P25, conclusions:

Conclusion 1: I think it is a bit of an open door. If units are delineated around areas which show most change it is logical that the patterns are more distinct than when you use a gridded approach. Conclusion 2: A large part of the variability is probably caused by differences in the energy balance and ablation regime, rather than precipitation

alone. Conclusion 3: See my earlier points. The stepwise increase seems to come after the lakes start to grow. Conclusion 4: ET depends not only on wind, but on humidity and radiation as well. Instead of the wind hypothesis an reduction of ET due to increased humidity is more plausible and this matches the increased in precipitation hypothesis.

References

Forsythe, N., Fowler, H. J., Li, X.-F., Blenkinsop, S. and Pritchard, D. (2017) 'Karakoram temperature and glacial melt driven by regional atmospheric circulation variability', Nature Climate Change, 7(August). doi: 10.1038/nclimate3361.

de Kok, R. J., Tuinenburg, O. A., Bonekamp, P. N. J. and Immerzeel, W. W. (2018) 'Irrigation as a potential driver for anomalous glacier behaviour in High Mountain Asia', Geophysical Research Letters, pp. 1–8. doi: 10.1002/2017GL076158.

---

## Referee Comment (RC2) · Anonymous Referee #2 · 3 Feb 2019

This paper presents an extensive study of glacier elevation changes and lake volume changes in High Mountain Asia (HMA) based on ICESat altimetry, and attempts to link the observed changes with climatic drivers, in particular precipitation. It builds on a number of related studies in the past, but takes a clear step forward by expanding the study region to the entire HMA and introducing a finer spatial zoning that accounts for orographic barriers and other known (and unknown!) reasons for regional patterns in glacier change. This provides some new insights to how HMA glaciers changed in the period 2003-2009, and makes it easier to link the findings with meteorological drivers and the observed lake growth within the endoheic basins of the Tibetan Plateau.

[Figure]

The authors employ a rather complex calculation and correction scheme that is sometimes hard to follow. I wish I had read the methodological appendices before trying to make sense of the shortened main text. This needs to be improved for readability. I do not even think it is needed to split up the text, because the appendices read well by themselves and have the same structure as the main text, without being overly detailed. There are also many repeated sentences between the two parts, which is annoying if you spend the effort to read both. The order of calculations and corrections is sometimes confusing, so I think that a few equations or a schematic would be helpful. A few of the corrections need to be better justified, especially since they also have the potential to introduce other types of errors (see the more detailed comments further down).

The authors claim (abstract and conclusion) to make a "spatially resolved estimate . . . of glacier volume changes for entire HMA", which would have been very useful since past ICESat studies have been spatially limited or based on older and less accurate versions of the Randolph Glacier Inventory (RGI). However, in the end, there is not a single glacier volume (or mass) change presented here, only figures of spatially averaged elevation trends for regions/zones that do not comply with past publications and RGI, making it impossible for the reader to make out the total numbers. Some aggregated numbers based on upscaling with RGI areas would be highly useful both for comparison with past studies (including GRACE) and as reference for glacier/climate assessments.

Despite these critical points, I do think that this study is highly valuable and should be considered for publication in the Cryosphere after careful revisions. I have listed a number of more specific comments, edits and suggestions in chronological order below. They refer to printed page and line numbers in the discussion manuscript which unfortunately often differ from true page-by-page line numbers.

P1, L2: A "diverse pattern" of volume change would be highly dependent on regional glacier area. I think you actually mean elevation changes in this context, so that should

be mentioned here or in the previous sentence.

P1, L3: I find it awkward to say "driven by . . . glacier sensitivity". The main physical driver is precipitation changes, but different glaciers can indeed have different sensitivities to that. I suggest to rewrite this sentence.

P1, L6: I think this statement is based on the reanalysis data which are discussed further down in the abstract. It is better to discuss topic-by-topic in a coherent manner.

P1, L13: "Considering evaporation loss,. . .". What do you mean? It sounds like you are not considering it here since you talk about "average annual precipitation". Please clarify.

P1, L16: Unclear what is meant by "geometry changes". Remove or explain.

P1, L18: Should be past tense, like the rest of the abstract, since it refers to a distinct period (2003-2008). Please check this elsewhere too although it is not a big issue.

P2, L2: Or the "Pamir-Karakoram anomaly" as suggested by Gardelle et al. (2013)

P2, L8: reduced/decreased evaporation (for consistency)

P2, L15-19: Some of these studies are not region-wide for HMA, but rather HKKH or Tibet only. That makes this study even more relevant (which could be highlighted).

P2, L29: Any reference(s) for the last two issues?

P2, L35: You concluded here or Kääb et al. concluded in 2015?

P3, L3-4: It is not intuitive what "hypsometries of individual years of ICESat samples" and "elevation trend in . . . sampling elevations" actually means. I think you should explain what hypsometry is and why it is important in this context, or use different wording to explain what you want to say.

P3, L7: 2018 -> 2008. And either you should explain what was special for this campaign or you should not mention it here.

P3, L15: Write out and reference RGI. And plural - regions of TP and Kunlun Shan?

P3, L20: I don't understand this sentence, and it doesn't seem needed either.

P3, L21: "The HMA glacier region is covered by..."

P4, L2: The fact that extensive parts of HMA has predominant spring/summer accumulation seems to contradict your reasoning to exclude all ICESat winter data (~March) because of variable winter snow, at least for some of your zones.

P5, L9: This is also nicely shown by Kraaijenbrink et al. [2017], but unfortunately hidden in the Supplementary information of the paper.

P5, L28: Reference for these data?

P6, L6: 1990s

P6, L12: See comment P4-L2. Considering that the ICESat data sampling is very limited, don't you miss out on a lot of potentially good data in TP and southeasterly regions where winters are relatively dry? I agree that the early summer data should be excluded though.

P6, L8: glacier samples

P6, L17: Any name(s)/reference(s) for the clustering methods you tested?

P6, L21. I think this is a good way to do it.

P6, L26: Reference or explanation for the four methods?

P6, L32: This paragraph is confusing and the correction needs to be better warranted. What is actually meant by "local reference elevation bias"? If a bias is truly local, then it is rather a local error (not systematic). In this case, would it not be better termed a "glacier-by-glacier bias"? But then comes the question of what causes such biases and why a correction is needed. Different DEM source date between glaciers within the same region is an obvious explanation in Treichler and Kääb (2016), but that is
less of an issue here since the SRTM DEM stems from a single year. Instead, a glacier-median dh correction might erroneously mute some of the real elevation change signal because the glacier-median time of ICESat samples will also vary from glacier to glacier.

P6, L33: Why just from "snow fall in the second part of the autumn 2008 campaign"? If I understand your correction right, the "glacier-by-glacier bias" is impacted by any type of elevation change between the time of SRTM and the respective ICESat measurements? I see the variation in glacier-median dh as a result of variable temporal and altitudinal sampling of ICESat between glaciers, as well as various errors in the SRTM DEM. If the latter is the main issue, why not use nearby land-samples to determine this correction?

P6, L35: What is the correction applied to? I understand it as a normalization of dh on a glacier-by-glacier basis by subtracting the median dh for each glacier, but this is not clear.

P7, L13: Do you actually mean non-glacier mass changes? Hence, removing the gravity signal from changing lakes to derive glacier mass changes from GRACE. Please clarify.

P7, L24: references?

P8, L5-7: Long and complicated sentence.

P8, L5-10: This section doesn't really describe a clear method beyond looking at the data and taking decadal averages. Is it needed? More confusing than clarifying.

P8, L13: 100 units? It doesn't look like so many.

P8, l17: What is done with those 34 units and why?

P8, L21: This correction appears out of nowhere. Remove or reference appendix B3.

P8, L24: Delete last sentence (already explained)

P8, L28: Interesting point, but since the grid cells are already overlapping by 50% and will be naturally smoothed by that, the conclusion is weak.

P10, L1: Nice!

P10, L35: I do not fully agree. See comments P6-L33 and P30-L20.

P12, L7: Any idea why?

Fig. 3: Interesting figure, but I suggest to use other colors for panel c to avoid confusion with the thickening-thinning colors in a-b.

P12, L10: Also mentioned in the caption. Once is enough.

P12, L15. I don't think this specification is needed.

Fig. 4b: Label regions according to Table 1.

Table 1: The caption is rather confusing, listing three time spans next to each other (belonging to different columns of the table) and giving volume change in unit mm without describing if it is per lake area or basin area.

Fig 6: The combination of two stations in the upper panel makes this figure unnecessarily hard to read. I suggest to split them in each their panel.

Fig. 7: Why does panel-a show ERA-Interim summer and panel-b MERRA-2 annual, and not either the same period for both products or both periods for one product. Also, the figure is only discussed very briefly, and well after Fig. 8 in the text. I think the figure is interesting, but to be included, it should be properly referenced and discussed in the text.

Fig. 8: The P-E curves appear are faded and hard to see, despite being most relevant in theory. I suggest you use a thicker line or sharper color/tone to improve visibility.

P16, L6. Specify these regions (NE, NW, C)

P16, L8: considering the high uncertainty; "results in" -> "suggests"

P16, L9: Fig. 8b?

P19, L20: . . .between the periods

P20, L23: "over 1988-2007" or "between 1988 and 2007"

P21, L18: This sentence is difficult to understand.

P21, L16-29: I would expect this paragraph to be closer linked with the interesting Fig. 3, as well as independent studies of velocity changes, of which the recently published study of Dehecq et al. [2019] seems particularly relevant.

P22, L13: Move authors out of the parentheses.

P22, L22: I think you really want to talk about elevation changes (or thinning) here since actual volume changes are so dependent on regional glacier area.

P22, L31. Uncelar sentence.

P22, L12: use abbreviated m w.e. a-1, as elsewhere

P22, L22: Is "glacier sensitivity to precipitation" an appropriate heading for this section? I feel it is more a discussion of orographic effects that cause different precipitation regimes on either side of a ridge, not really whether glaciers are more or less sensitive to precipitation in general. Or you need to better explain what you mean by "sensitivity".

P23, L30: . . .both surging glaciers and glaciers recovering. . .

P23, L32: Combine references with same authors.

P23, L8: This paragraph is very detailed compared to the others and could be shortened.

P25, L20: The Conclusions section provides a good summary, but would benefit from a shortening to better highlight the main findings and outlook.

P25, L21: This study has many new and interesting aspects, but it is not the first

one and does not actually present any volume changes. I would rather highlight the improved zoning and joint analysis of lake changes as the most unique part of this paper.

P25, L11: The selective mention of MERRA-2 (which fits with your observations) and not ERA-Interim (which doesn't fit) is peculiar as long as you cannot identify reasons why one product should be better than the other. Mention both products or none.

P26, L15: Is this significant? If not, no need to mention as a conclusion.

P26, L10: Since Cryosphere has easy support for auxiliary data, it would be very nice for the community if the zoning (including agacier area and averaged dh/dt) is provided with the paper, not only on personal request.

P28, L18: ICESat period

P30, L18: Method A is not clear.

P30, L20: Doesn't the B correction correction also introduce an error due to ICESat's variable temporal sampling? I.e., if a glacier is thinning, then ICESat observations in the later years of the mission would naturally have lower dh values than expected from the general dh-elevation trend. This is the same issue as pointed out for the cG correction (see comment P6-L32). Are both corrections applied in the case of method B?

P30, L21: What is meant by "filtering"? If you mean removing/culling data, then useful observations would also be removed and I see no reason for that as long as you can rather introduce a weighting scheme like Method D.

P30, L10: Thanks, this shows that you are aware of my issue with the cG correction and method B. Since it is in the end only applied to 6 units – is it really needed? And what about the same issue for Method B?

P32, L24: Standard error of the mean?

**References**

Dehecq, A., et al. (2019), Twenty-first century glacier slowdown driven by mass loss in High Mountain Asia, Nat. Geosci., 12(1), 22-27.

Kraaijenbrink, P. D. A., M. F. P. Bierkens, A. F. Lutz, and W. W. Immerzeel (2017), Impact of a global temperature rise of 1.5 degrees Celsius on Asia's glaciers, Nature, 549, 257.

---

## Author Comment (AC1) · 15 Apr 2019

**Response to referee comments**

**High Mountain Asia glacier elevation trends 2003–2008, lake volume changes 1990–2015, and their relation to precipitation changes**

Désirée Treichler, Andreas Kääb, Nadine Salzmann, and Chong-Yu Xu
The Cryosphere Discussion, doi: 10.5194/tc-2018-238

We would like to thank the two reviewers for their constructive feedback and valuable input that certainly helped to improve the article. Detailed responses are provided below, together with a mark-up manuscript version where the changes made in response to the referees' comments are highlighted.

**Anonymous Referee #1**

*General*

*The authors present an interesting study and they analyze surface elevation changes in High Mountain Asia with ICESat data between 2003 and 2008. They hypothesize that the positive glacier mass balances found in the eastern Pamir, Kunlun Shan and the central TP can be explained by a step-wise increase in precipitation. They approximate the precipitation change by quantifying changes in lake volume of endorheic lakes, station and reanalysis data. I believe the study definitely has scope to be published in the cryosphere, but I find that the conclusions drawn are too strong and are not supported well enough by what the (uncertain) data shows. I have identified the following issues that need to be addressed before the paper is acceptable for publication:*

*1. Previous work has aggregated surface elevation changes on a 1 degree or 2 degree grid. In the present study the authors have made their own delineation, which they acknowledge to be subjective. The procedure for delineating the spatial units is not clearly described (p6, l17-24). It comes across as if polygons are drawn around region where trends are most clear and obviously the resulting zonal map (Fig. 2A) looks better than the gridded map (Fig. 2B). The use of the zones needs better justification and they have to be objectively defined ideally without prior knowledge about the ICESat trends.*

It is unfortunate that our explanation of spatial unit delineation came across subjective or even unsound. In contrary to what the reviewer seems to assume, our zonation did not make our life easier (i.e. we were not tuning the units to make the results look great) but we rather spent a considerable amount of time to ensure the spatial aggregation is as appropriate as possible. Using a gridded approach or the RGI regions would have been straight forward, but unfortunately these spatial zonations to some degree violate the important principle that in a classification, samples within one group should be maximally similar – and maximally dissimilar to other groups. Existing spatial groupings (including the RGI regions, which were drawn to split a lot of glacier vector data into smaller chunks of approximately equal disk space) or regular spatial grids have several issues: they split mountain ridges into several regions without there being any topographic/climatologic/ elevation reason to do so, they merge several orographic mountain ridges into one unit (the eye prefers roundish units) even though climatic/orographic conditions and elevations change very quickly across sequences of mountain ranges, and they may even split individual glaciers into several spatial units.

We tried to make our zonation as objective as possible by analysing topography and glacier statistics (sizes, types, mass balances, elevations, slopes, aspects…) within each unit, and consulted experts as well as literature. The most objective approach would have been to derive a spatial grouping from ICESat samples directly (using the above glacier statistics of the samples), but our efforts to establish such an automated clustering were not successful; we quickly realised that designing a model rule set would become much too complex. We indeed used ICESat trends iteratively, but only to check whether already drawn units yield robust (and reasonable) glacier surface change rates. If not, we merged units, or in some cases also split units if it seemed like we were capturing a mixed signal of glacier mass balance evolution.

We rewrote the methods paragraph and Appendix in question to better justify the zonation process and did our best to emphasise that we aimed for a transparent and objective approach, as far as this was possible.

*2. The lake changes are solely attributed to precipitation changes and I have some doubts about this assumption. I think a potential important factor can be the change in permafrost. Much water is stored in frozen form in the soil. An increase in the active layer as a result of rising temperatures may also considerably impact the lake water balance. However this is not at all discussed, and temperature trends are not mentioned either. Therefore I recommend to include references to changes in permafrost hydrology and to quantify spatially also the temperature trends based on the reanalysis datasets.*

It was not our intention to attribute lake changes to precipitation changes only (although we believe they are the main driver). We are aware that in particular evaporation might be an important factor, not least due to strong warming trends and other climatic/meteorological changes. For example, Zhang et al. (2018) suggest lake growth may partly be explained by a significant decrease in evaporation during the past 30 years. – However, we did not discuss thawing permafrost, as the reviewer correctly points out.

The question of how much water may have been released due to thawing permafrost is a difficult one, also for other regions of the world that are better studied than the TP. We discussed thawing permafrost as a potential source of water with experts within our research groups. According to S. Westermann (personal communication), this strongly depends on how much ice there was in the ground in the first place (and on what is replacing the melted ice), and this is largely unknown for the TP. In principle, an increase in the active layer will also allow for more water storage (which again prolongs/increases the runoff to rivers or, in our case, lakes). This however requires that there is sufficient water available to fill the (newly) available storage – from precipitation or potentially also possibly snow melt (only where the ground is thawed during snow melt). Thus, the parameters needed to quantify runoff from thawing permafrost (or groundwater storage of previously frozen ground) are largely unknown. The spatial extent of increasing active layer depths and thawing permafrost on the TP is poorly known due to lack of measurements but has been the
focus of several older and recent modelling studies. Discontinuous/sporadic permafrost can be found
everywhere on the TP, continuous permafrost is found in the northern half (our regions: NW, NE,
Central and upper half of E TP; see e.g. Zou et al, 2017; Ran et al, 2018). Recent and ongoing
temperature rise lead to an increase in the active layer and degrading permafrost (Ran et al, 2018).
There are however also other ongoing processes that protect the TP such as desertification which
leads to cooling of the ground (Xie et al, 2015).In general there seems to be agreement that
permafrost degradation was greatest in the southern (where we find little lake change / lake growth)
and eastern parts of the TP (strong lake growth).
Considering the complexity of the effects of temperature change on the cryosphere (glaciers,
permafrost) and atmosphere (e.g. evaporation), we think including also temperature data in the
analysis would be too far off topic for this study and would only result in duplication of existing work
with an actual focus on temperature trends. Instead, we state the ongoing warming trend both in
introduction and discussion by citing relevant references and believe it can be assumed that the
ongoing temperature increase is generally known.
In the revised manuscript, we emphasize in the text that the TP is undergoing substantial
warming that rather exceeds warming trends elsewhere in the world. We emphasize the potential
role of changes in evaporation and added permafrost thawing as another possible contributor to lake
volume changes, with references to recent studies.

*Furthermore the assumption that most lakes are endorheic is quite strong. The subjective description*
*of how the HydroShed dataset has been manually modified is a bit worrying (p7, l31-34). It would be*
*recommendable to clarify this.*

The reviewer is right in that there might be groundwater exchange between some lakes, and
we mention this in the first paragraph of section 3.3. The inner TP and thus the catchments we
analyse, however, are endorheic. Groundwater exchange between lakes within (or even between)
endorheic catchments does thus not affect our estimate of water volume change over larger spatial
areas. We added a note regarding this in sect. 5.1.
As the HydroSHEDS dataset was created at 15 arcsec resolution it does not everywhere produce
correct endorheic catchments (Lehner et al., 2008). We used better resolved SRTM DEM data (thus
mainly using topography as a definition for endorheic basins, as subsurface water flows are unknown)
and time series of Landsat imagery (to detect surface water flow and ensure our catchments
correctly reflect where lakes split/grew together/emerged since SRTM DEM (and HydroSHEDS)
production) to adjust catchment borders where they were incorrect.
We rewrote the paragraph to clarify why and how the HydroSHED dataset has been modified.

*3.  MERRA-2 is a reanalysis dataset which is known not to perform very well in high mountain Asia,*

We wonder why the reviewer suggests MERRA-2 to perform badly in high mountain and
would be interested in according references. We agree that all reanalysis products (and other
precipitation estimates) have issues in performance, in particular at high elevation (Reichle, 2017;
Sun et al. 2018), but MERRA-2 typically shows among best performance, particularly also for our
region. Chen et al. (2019) assessed CFSR, ERA-Interim, JRA-55, MERRA-2, NCEP-2 and found for
instance that precipitation and drought characteristics are best represented by MERRA-2 across
China. And among all the five sub-periods they analysed, monthly drought areas and severity
obtained from MERRA-2 in 2001–2007 agree best with that obtained from the observed data in both eastern and western China. Moreover, also Cuo and Zhang (2018) have chosen ERA-Interim
and MERRA-2 for their study on spatial patterns of wet season precipitation vertical gradients on the
Tibetan plateau and the surroundings.

*yet strong conclusions are drawn based on the projected changes in precipitation. The results of*
*ERA-Interim are largely ignored since they not match as well to the observed lake and glacier*
*changes. It is recommended to better justify the use of MERRA-2 and show a comparison with the*
*station data or provide another argumentation why this reanalysis dataset should be used. It may also*
*be worthwhile to use the recently released ERA5 dataset which is the high resolution successor of*
*ERA-Interim.*

The reviewer is right in that our discussion mainly focused on MERRA-2 data. We fixed this
and now consider ERA Interim data better in results, discussion and conclusions.
We added the above mentioned references to recent studies which justify (and assess) ERA Interim
and MERRA-2 data for the TP to section 3.1.
After discussions with experts for reanalysis products on the TP (see Orsolini et al, 2019), we don't
think that ERA5 is a useful choice to model precipitation in HMA as this reanalysis does not contain
any data assimilations on snow cover for pixels above 1500 m: ERA5 is produced using the data
assimilation in CY41R2 of ECMWF's Integrated Forecast System IFS, where the use of satellite data
snow extent is switched off for altitudes > 1500m (ECMWF, 2016; see also point T5 of known IFS
forecasting issues on the ECMWF wiki:
https://confluence.ecmwf.int/pages/viewpage.action?pageId=28328424). This means that satellite-
based information on snow cover fraction is not used in our entire area of interest, resulting in a
poorer amount of real data forcing for in particular precipitation modelling in ERA5. Over the TP, also
no station data is used. Orsolini et al (2019) thus find that despite its lower spatial resolution, the
older ERAInterim is more appropriate and accurate in HMA. Additionally, ERA5 data prior to the year
2000 only became available in January 2019, long after this paper was submitted, and the reanalysis
data are currently being moved from ECMWF to the Copernicus Climate Data Store – and monthly
means of daily means are not (yet) available via the CDS API. Given the circumstances we
unfortunately didn't succeed to verify the datasets assumingly poorer performance by creating the
same maps as for MERRA-2 and ERA Interim for this response letter, although we also were curious
about this.
However, we added a reference to ERA5 for potential future studies in section 5.2.

*4. The use of actual evapotranspiration from MERRA-2 to derive a lake basin water balance is*
*questionable and highly uncertain. The uncertainty needs to be discussed and quantified or ideally an*
*ensemble of re-analysis products should be used. The authors may even consider leaving out the*
*whole reanalysis part given its uncertainty. Linking the lake and glacier dynamics is already exciting*
*enough.*

We agree that reanalysis products have high uncertainties in data sparse regions such as
HMA. We would however like to emphasize that we are not aiming at providing accurate numbers
but rather a rough estimate. In that sense this article is a stub that hopefully leads to further
integrated studies across the traditional research disciplines. Reanalysis products are an important
source of information for investigating the climate in data sparse regions. We therefore think
including the reanalysis part is valuable for this paper and prefer not to leave it out. However, we
doubt that a full-scale ensemble analysis would yield much different results within the scope of this paper. Reanalysis comparison studies have been done for this region (e.g. Cuo and Zhang, 2018; see comment above) and justify the two products we have chosen.

In the reviewed version of this manuscript, we better acknowledge and discuss uncertainties associated with reanalysis products (section 5.2) and added uncertainties (propagated error) to the evapotranspiration numbers in sect. 4.4.

*5. The authors conclude that the lake changes match the glacier surface elevation change very well. I am not sure if I agree. Table 1 shows positive lake volume changes everywhere, while the glacier mass balance shows contrasting trends across the region. In addition the periods do not match (1990-2000 versus 2003-2008). The same holds for Figure 4. I do not see many similar patterns between Figure 2A and Figure 4. The lake growth very clearly starts in 1995 (Fig 5), but the increase in precipitation occurs about 5 years later (Fig. 6 and 8), so that does not make sense to me.*

Even with additional precipitation, glaciers may still melt in a warming climate, but volume changes might be slowed down and glaciers might adjust their geometry due to dynamical changes: Increased precipitation causes more input (at high elevations) while increasing temperatures might enhance melt (mainly on the tongues). Such a change in climatic regime will cause the glacier to shrink but thicken – with some delay (due to ice flowing slowly). We find indications that exactly this is happening on the TP (fig. 3 in the article). Thus, the patterns of glacier and lake changes do not need to have the same sign to match.

The reviewer is right in that the studied time periods don't match exactly. However, this does not affect the outcome of this study. Unfortunately, the lack of data, and uncertainty in the available data, prevents us from studying shorter time periods: ICESat data are only available during 2003-2008, and uncertainties in precipitation/reanalysis data as well as lake data (Landsat imagery) in particular prior to the year 2000 require temporal summaries. We chose decades due to three reasons:

1) the exact date of the precipitation increase is not clear. From visual inspection of precipitation time series (figs. 6 and 8), the increase started/happened somewhere between 1995 and 2000 – ERA Interim rather suggests the former, and MERRA-2 and most stations the latter.

2) the exact date of the lake increase is not sure either due to data scarcity in particular between 1995 and 2000. Note that the data in fig. 5 in the article is filtered with a 7 years window (before computing the median) which contributes to shifting the onset of volume change in the middle of the period of question. Figure 1 below shows regional median time series from unfiltered data – and there, e.g., the lakes in the region northeast seem to start to increase in 1999. Time series of large individual lakes (e.g. Zhang et al, 2017;2018; Song et al, 2015) show different onset times but mostly closer to the year 2000 than 1995.

3) the (shorter) ICESat period lies in the middle of the 00s decade. By comparing the two we assume the 2003-2008 glacier signal is representative for the entire decade. While glacier mass balances vary annually and are an immediate feedback to precipitation input and melt, the glacier's geometry may take some time to adjust (see comment above).

Considering that, decadal averages and differences are useful measures for the scope of this study.

In the revised manuscript, we better link lake/precipitation changes with the glacier signals/glacier geometry adjustment visible in fig. 3 (section 5.3).

[Figure]

**Figure 1: corresponds to Figure 5 in the paper, but regional medians are computed from non-filtered lake time series.**

*6. Glacier changes are explained only from an accumulation (and precipitation change) perspective.*
*However a glacier mass balance is the results of accumulation and ablation. Total precipitation may*
*increase, but if the temperature increases as well that may still result in less snow. In addition an*
*increase in temperature may also enhance the melt and other energy balance terms may change.*
*Recent work explains the Karakoram anomaly as a result of more summer snow fall and less melt due*
*to less incoming shortwave radiation due to more clouds and a higher albedo (de Kok et al., 2018).*
*An- other important study identified the Karakoram Vortex, which draws cold air into specific part of*
*the region (Forsythe et al., 2017). None of these factors are considered.*

We completely agree with the reviewer, glacier mass balance is not governed by
precipitation alone. It is unfortunate that this does not come across well enough in the manuscript.
As stated above, we are very aware of changes in glacier climatic conditions: changes in both
precipitation and temperatures may cause glacier geometry changes. The ICESat results indicate this
is happening on the TP (fig. 3).
We added the two suggested references that offer explanations on why precipitation may
have increased in parts of the study area. We also state more obviously that temperatures are rising
in HMA and discuss the effect of coincident precipitation and temperature changes on glaciers on the
TP in more detail (section 5.2, 5.3).

*Specific comments*

*P2, l25: I recommend a more detailed comparison with the Brun et al., (2017) results*

This is done in that study already. We added a reference in the text.

*P3, l6-7: HMA does not have a typical winter snow fall – summer melt cycle. While this may be true in*
*the west, in the monsoon dominated areas the winters are generally dry and there is synchronous*
*ablation and accumulation. Therefore (high altitude) summer snow events may also cause a bias. In*

*addition I wonder given the type of trend analysis used, why a single anomalous event in autumn 2018 causes such a bias.*

That's correct; summer snow fall might cause bias in the early summer ICESat campaigns. These are excluded (there are only three years with late spring/early summer campaigns). ICESat autumn campaigns are the only ones where it is dry everywhere. Although there might be some regions where winter campaigns contain useful data (namely the TP, although detailed analyses not included in this paper show evidence of winter snow fall from ICESat data also in this area), we prefer to use a consistent approach throughout the entire study area. Winter campaigns may though be used for local studies focusing on small regions. (See also the reply to comment P6, L12 of reviewer 2.)

The sentence in question is thus a statement that is true in general, not for HMA specifically/only. Precipitation/accumulation patterns for HMA glaciers are described in section 2.

We agree that it might seem strange that the December 2018 snow fall has such a large influence on the fitted trend. That's however not a peculiarity of our study; data points in each end of a trend analysis have a considerably larger influence on trend slopes than other data points. For a discussion/explanation see Appendix D in the paper and our reply to comment P12, L7 of reviewer 2.

*P4, l5: What is meant by precipitation availability? Just precipitation is enough I would suggest.*
CHANGED

*P5, l13: insert here a paragraph on ablation and radiation regimes across HMA?*

The focus of this study lies on (changes in) accumulation. However, to better acknowledge other ongoing changes (temperature rise, changes in circulation patterns and effects thereof), we integrated information about glacier ablation (and changes thereof) in the paragraph.

*P5, l20-25: not sure if it adds value to mention what has not been used. It is absolutely fine to use SRTM.*

Our concerns with temporal inconsistency and data voids of other DEMs available in the area might be useful for readers not so familiar with the different DEMs. We moved the information to the Appendix.

*P5, l29: some validation of MERRA5 is required. Large cold biases in reanalysis datasets are very common and this may have very large effects on the modelled snow- fall for example.*

Reanalysis products are expected to be more uncertain in data sparse regions like HMA, but they are also an important source of information (the alternative is no data). Reichle et al (2017, 2017a) did rigorous analyses of the MERRA-2 performance in particular also on snow and show that the product is doing very well there. A separate analysis/validation of MERRA-2 is beyond the scope of this study.

To justify the use of the chosen reanalysis products, we added a short note to section 3.1 and further references to Appendix 3. (We also moved some of the details on reanalysis products in section 3.1 to the appendix, to shorten the rather long paragraph and to better match the primary amount of information given for the other data products.)

*P5, l4: add reference to the Global Surface Water dataset* DONE

 *P6, l14-16: why use three different methods and then use the average? This assumes each method*
*performs equally well. Are there no arguments why a certain method is preferred in this case?*
*P6, l16: Same. Why use four different ways of hypsometry corrections and then take the average?*

The three regression methods that we use are supposed to be little affected by outliers. As
the reviewer mentions in his comment to P3, l6-7, and we mention in Appendix D in the paper, it is
surprising that a small amount of samples (December 2008 campaign) affected by a systematic offset
has such a large influence on linear fits. Treichler & Kääb (2016) tested a t-fit as an alternative to the
previously used robust regression for ICESat data analyses and found no difference in their
performance/accuracy. In this study we additionally added the non-parametric Theil-Sen linear
regression that is commonly used e.g. in hydrological analyses and should fit our data situation well.
However, despite having 100 spatial units at hand to compare the three fits, we find that none is
systematically performing 'better' (i.e. visually less sensitive to outliers, systematically different than
the other two etc.). The trend slope differences between the three fits are within 0.1 m/a, which is
well within the error estimates. We thus chose to use the average of the three.
The situation is similar for the hypsometry correction methods. All have the same goal, but
the approach is somewhat different. When looking at their individual performance for single spatial
units, we find that some corrections sometimes perform 'worse' than other due to the nature of the
local sample distribution in time and space (i.e. they don't fully correct for bias from hypsometric
sampling, that is why we came up with several correction methods in the first place). Mostly,
however, at least three out of four corrections result in trend slope differences of within 0.1 m/a. For
local studies, it would be possible to choose a single correction method and argue why it is best. For
the entire study area, however, this is not possible. By using the average of four methods we ensure
a consistent approach that maximises the accuracy of hypsometric correction (i.e. minimises a
potential error introduced if one of the methods over- or undercorrects hypsometric bias).

*P7, l4: Snow does not fall only in winter in HMA, so how are other campaigns influenced?*

Late spring/early summer campaigns have little data due to cloud cover in large parts of
HMA. In many of our spatial units, we see a snow signal for winter campaigns (in average less
negative / more positive dh compared to autumn campaigns; in some of our units and if larger
regions are analysed, the snow on-off signal is even visible annually – this depends on the existence
of other vertical biases). We added a small side note to the first paragraph of section 3.2 to explain
the choice of using autumn campaigns only more explicit: "*We used only samples from ICESat's 2003-*
*-2008 autumn campaigns,* the season with least snow cover in entire HMA*, to avoid bias from*
*temporal variations in snow depths (see introduction)."*

*P7, l17-20: Again why multiple methods?*

The two datasets used have different strengths and weaknesses: ICESat-derived lake surface
elevations are far more accurate but available only for about a tenth of all lakes. SRTM elevations
have uncertainties of several metres (see e.g. Treichler and Kääb, 2017) but are available for almost
all lakes. Using the two complementary methods gives us the possibility to combine their strengths
and to validate them against each other, as we write in the end of the paragraph. We added a
sentence earlier in the paragraph to already there show the advantages of using two complementary
methods.

*P7, l20-24: If most lakes are growing and the reference DEM is SRTM (~2001) or IceSat (2003-*
*2009) then how is the water volume change reconstructed prior to this period as there is no*
*information about the lake bed elevation below the water. A discussion regarding the*
*uncertainty of using an "above the water" volume-area scaling would be useful.*

We are aware of uncertainties associated with "above the water" lake level-area scaling (and
other uncertainties in our data) and ran several control runs of our analysis with maximally
conservative assumptions (not included in the paper. For an example, see Figure 2 - apologies for the
complexity, this was used for quality control during analyses and originally not meant for publishing).
The vast amount of data and large number of lakes forced us to use the same assumptions and fits
for all lakes. For a single lake, this indeed causes great uncertainties – some assumptions will lead to
overestimation, some to underestimation of the actual lake volume changes (dV, see below).
Summarised over the entire study area, however, errors can be assumed to mostly average out. We
believe that the effects of "above the water" lake level-area scaling and other uncertainties are of
comparatively little importance for the scope of this study, which aims at reconciling changes of
glaciers, lakes and precipitation. In other words, we don't aim at exact estimates for single lakes but
rather a summarised approximate estimate of dV and their spatial distribution.
SRTM: data for the SRTM DEM was acquired in February 2000. Lakes on the TP have seasonal
cycles in line with precipitation and are usually largest at the end of the summer. We use annual
maximal lake extents in our analysis – with a reference DEM from winter 2000 where the lakes were
smaller (and likely covered by ice) we can thus go somewhat beyond that date (assuming lake growth
started before 2000). It is however true that SRTM lake level elevations prior to 2000 have higher
uncertainty: Figure 2 below shows that the SRTM data and data points of the 90s have higher
associated uncertainties (the red area-lake level scatter cloud has a higher spread; time series data
points <2000 are less linear). The effect of extracting SRTM lake elevations for lake areas smaller than
during SRTM data acquisition is that pre-2000 SRTM lake levels may be too high, resulting in too
small dV (see orange arrows in Figure 2). Comparing SRTM and ICESat time series in Figure 2, this
might be true and could be one of the reasons why dV from SRTM data are smaller than for ICESat
data.
ICESat: choosing a linear area-lake level fit essentially assumes a parabolic bathymetry
between minimum and maximum lake area. To find the best fit, we analysed area-lake level
relationships using all 18 ICESat campaigns (and corresponding maximum lake area of the ICESat
acquisition month, plus one month before and after, to minimise data gaps). We found no obvious
indications that a different fit would be more appropriate – for above-water data points only, though.
Assuming a constant shore slope instead, which is a plausible alternative to our implicit assumption,
the fit would have to be done with sqrt(A). The orange arc and arrows in Figure 2 show in a
qualitative way how this would affect the computed dV for randomly chosen Ayakkum Lake:
computed lake levels for small areas would be lower, causing a greater dV between the 90s (small
lake areas) and later dates. In that example, our estimate is thus rather under- than overestimating
dV. However, the error from the assumed bathymetry underlying the chosen fit has comparatively
little influence on dV: in numbers, the area change is much greater than the change in lake level, thus
errors in the assumed lake area would cause more bias than it is the case for lake levels (see our
reply to comment P12, l7-8 below). As one control approach (not included in the paper) we thus
computed lake volume changes also for potential maximal lake areas – assuming all NaN cells in the
global water dataset within our lake masks were water cells, which is very likely overestimating the actual area by far. Propagating this in the analysis leads to smaller dV, shown as cyan/magenta time
series in panel 4 in Figure 2.
In the appendix, we now explicitly mention uncertainties associated with "above the water"
volume-area scaling and SRTM lake elevations for lake areas that are smaller than during SRTM data
acquisition.

[Figure]

**Figure 2: Changes of Ayakkum Lake, East Kunlun Shan. Left: Area-lake level scaling using data points with >95% data. Second panel: lake level vs time, third panel: water area vs. time, right: lake volume change vs. time (using the data from panels 2 and 3). Orange: How a bathymetry with constant slope would change fitted lake levels and computed volumes.** Colours and markers: Circles marked with +/x denote data points with <5% no data cells within maximum lake extent; decadal averages (horizontal bars in panels 2-4) are only computed from these. Red/green bars mark the potential maximal area of data points if all no data cells were counted as lake area. Blue – ICESat data; black – ICESat data extrapolated for area data points (years) without lake level data, using the linear fit through blue x; red – SRTM, green – area (Landsat). Cyan/magenta: most conservative alternative that uses the potential maximal lake area instead (for ICESat, this also changes the extrapolated lake levels: circles in cyan).

[Figure]

Ayakkum Lake, width ~20km. Screenshot from Google maps

*P8, l5-10: the authors indicate that the reanalysis data is not accurate, but still it is used to draw*
*strong conclusions.*

In the revised manuscript, we better consider the uncertainty associated with reanalysis data
in the discussion and conclusions. We removed that paragraph and rather added some information on the representativeness and uncertainties of the data to sections 4.4 and 5.2, where they better fit
the text flow.

*P12, l7-8: Why is the ICESAT based lake level change 1.55 times as large? Does that point towards a*
*systematic difference between SRTM and ICESat in off-glacier areas?*

Potential systematic vertical offsets between the two datasets don't influence the analysis,
as we look at changes in lake level elevations (for each dataset separately) rather than absolute
surface elevations. However, SRTM elevation accuracy is at least a magnitude lower than ICESat
elevation accuracy over (flat) lake surfaces (see Treichler & Kääb, 2017, for a discussion of elevation
accuracy of the two datasets in mountainous terrain). A potential explanation for the DEM elevation
uncertainties and variability of extracted DEM elevations at the lake shores (appendix C1) is
penetration of C-band radar into sandy ground (Williams and Greeley, 2004) that varies spatially
depending on moisture content (data acquisition was in February 2000, and local
conditions/temperatures during acquisitions are unknown). Additionally, the processing steps to
mask/interpolate water-covered areas (below lake ice) without radar backscatter during DEM
production are not known, resulting in greater uncertainties for DEM cells that correspond to the
February 2000 lake shore.
As the reviewer pointed out above, pre-2000 SRTM lake surface elevations are more uncertain, they
may be too high for years when the lakes were smaller than during February 2000. As mentioned in
our reply above, this might explain why results using SRTM data yield smaller numbers than for
ICESat data. Note, however, that the difference between the two estimates is considerably smaller
for dV (1.09 times), as the influence of areal changes (up to many km$^2$) is much greater than the one
of different lake level estimates (centimeter-meter).
We added a short explanation and reference to appendix C1 to the text.

*P12, l8: lake growth = water level increase?*     CHANGED

*P13: Fig. 4: What is meant by median lake area? Express 4b as mm/year to make it comparable to*
*precipitation rates?*

We mean the median lake areas from the 1990--2015 annual lake extents. We changed the
text in the figure caption to plural to make it clearer and added an explanation to appendix C on what
we mean with median lake areas.
We understand the reviewer's suggestion, but prefer to keep the plot units as is (the same is
true for the numbers in table 1). Precipitation is expressed in mm/a, thus a precipitation increase
(between decadal averages) also has the unit mm/a. The precipitation data used in this study
suggests that the nature of this increase was rather step-wise than gradual. Lake volume changes (dV)
are computed as a volume change between decadal means and may either be expressed in total
volume change or possibly, as the author suggests, in mm/a to have comparable units. However,
spreading the lake increase evenly across ten years might not be correct (fig. 5 in the article), as it
looks like most of the extra lake volume appeared within a rather short time (between ca. 1997 and
2001). Thereafter, lakes continue to grow, but more slowly. Considering a step-increase in
precipitation, the story would then be: lake volumes increased quickly within only few years, and
that rate of growth is directly scalable to the extra precipitation. After the initial increase, with
ongoing higher precipitation rates, lake growth quickly decreases as the lakes approach a new
equilibrium. Thus, it might thus be equally correct to distribute dV between e.g. five years. We prefer to leave it up to the reader to decide what is most appropriate (as we write on page 14 in the
manuscript).

*P14, Figure 5: Very interesting to see the abrupt increase from 1995 except for the Qilian Shan*
*region.*

Note that lake time series are median-filtered due to data scarcity for the years 1995--1999.
There is thus some uncertainty on the exact timing of the onset of lake growth: For lakes with large
data gaps, the filter has a tendency to place the onset of lake growth in the middle between 1995
and 2000. See also our reply to the reviewer's comment 5) above.
We added a note of caution in the text (section 4.3) explaining the above.

*P15: Instead of the data in table 1 I suggest to sync the periods and show the 2003- 2008 glacier*
*mass balance, lake volume change, re-analysis precipitation change and re-analysis precipitation*
*minus evapotranspiration change.*

See our reply to the reviewer's comments 5) and P13: Fig.  4. above.

*P16. Fig6:  Add the reanalysis data for the same pixel as the stations to assess its validity?  Stepwise*
*increase (if significant) occurs around 2000 which is 5 years later than the lake increase.  Same for*
*Fig.  8.  One solution could be to look at trends and test their significance rather than focusing on the*
*"step-wise" increase.*

We acknowledge the reviewer's concerns but don't think a comparison between station data
and the corresponding reanalysis grid cell will be useful within the scope of this study. Kääb et al
(2018) did extensive tests of various meteorological data to model the Aru glacier collapse on the
western TP. They assessed both the Shiquanhe station data (closest) and the NCEP-NCAR, MERRA-2,
ERA-Interim and HAR reanalyses (supplement). They found that the different data sets differ
substantially especially in terms of precipitation amounts (fig. 7a), and that  "Shiquanhe
meteorological data gives unreasonably warm temperatures (…) either (due to) a particularly 'warm'
setting of the meteorological station, or an applied lapse rate that is not representative for the
region". They used ERA-Interim data where especially precipitation had to be heavily corrected to
receive required input amounts for glacier modelling. The corrected precipitation shows a step-like
increase around 1997, rather than a trend.
Concerning the timing of the postulated step-wise increase, we refer to our reply to the
reviewer's comment 5.
It is thus on purpose that we chose decadal means rather than trends, even though trends
are very popular within climate analysis and their use is rarely questioned. However, especially for
short periods (e.g. a decade), trends are extremely sensitive to the "end years" (same as for our
ICESat glacier time series – as the reviewer pointed out above; see Appendix D and our reply to
comment P12, L7 of reviewer 2 for an explanation). Naturally, precipitation values vary greatly from
year to year. We are confident that decadal means are more robust and better suited for the
purpose of, and data used in this study.
A side note – it might well be that the precipitation increase did not occur at the same time
everywhere on the TP. After all, this is a pretty large area and circulation patterns (and changes
thereof) are not the same everywhere – and both the literature cited by the reviewer and in our
response letter suggests that various changes are happening on the TP.  Looking at this in more detail would be very interesting for a follow-up study focusing on precipitation and circulation data rather
than glaciers and lakes.

*P19 l30-31: very thin basis for this conclusion.*

We rephrased the sentence and included also a reference to fig. 3a which shows thickening
of the upper 50% of glacier area in the regions of question.

*P22, paragraph 5.3: very interesting finding that the southern slopes have less negative mass*
*balances. It seems to be related to a higher mass turnover and a reduced sensitivity of the mass*
*balance to temperature changes.*

Note that the first orographic ridge, and thus our spatial units, also include the northern
slopes on that ridge (very few glaciers face south). We now better stress that information in the
appendix.

*P25, conclusions:*

*Conclusion 1: I think it is a bit of an open door. If units are delineated around areas which show*
*most change it is logical that the patterns are more distinct than when you use a gridded approach.*

We refer to our reply to the reviewer's first comment.
However, "…units are delineated around areas which show most change…" – this is
essentially what we want to achieve! As long as the data points within the spatial unit are
representative for the glaciers within that unit (which we assessed carefully), these units will
emphasize local differences (and thus, eventually, help to understand why these glaciers behaved
differently) rather than blurring that signal.

*Conclusion 2: A large part of the variability is probably caused by differences in the energy*
*balance and ablation regime, rather than precipitation alone.*

We find that the spatial distribution of precipitation changes and glacier elevation match the
spatial variability of glacier changes well. This does of course not exclude that other factors are
affected, too. Energy balance and ablation are influenced by both precipitation changes and
underlying temperature trends. We hope that the readers will understand this from our revised
manuscript, where we better stressed other changes happening in the region (especially the
temperature rise).

*Conclusion 3: See my earlier points. The stepwise increase seems to come after the lakes start to*
*grow.*

We refer to the replies to the reviewer's earlier points, it is difficult to pinpoint an exact date
of when the changes happened.

*Conclusion 4: ET depends not only on wind, but on humidity and radiation as well. Instead of*
*the wind hypothesis an reduction of ET due to increased humidity is more plausible and this matches*
*the increased in precipitation hypothesis.*

We agree and added increased humidity as a potential cause for reduced evapotranspiration
(also in the discussion).

**Anonymous Referee #2**

*This paper presents an extensive study of glacier elevation changes and lake volume changes in High Mountain Asia (HMA) based on ICESat altimetry, and attempts to link the observed changes with climatic drivers, in particular precipitation. It builds on a number of related studies in the past, but takes a clear step forward by expanding the study region to the entire HMA and introducing a finer spatial zoning that accounts for orographic barriers and other known (and unknown!) reasons for regional patterns in glacier change. This provides some new insights to how HMA glaciers changed in the period 2003-2009, and makes it easier to link the findings with meteorological drivers and the observed lake growth within the endorheic basins of the Tibetan Plateau.*

*The authors employ a rather complex calculation and correction scheme that is some-times hard to follow. I wish I had read the methodological appendices before trying to make sense of the shortened main text. This needs to be improved for readability. I do not even think it is needed to split up the text, because the appendices read well by themselves and have the same structure as the main text, without being overly detailed. There are also many repeated sentences between the two parts, which is annoying if you spend the effort to read both. The order of calculations and corrections is sometimes confusing, so I think that a few equations or a schematic would be helpful. A few of the corrections need to be better justified, especially since they also have the potential to introduce other types of errors (see the more detailed comments further down).*

The methods section and appendices have been rewritten for better readability and clarification.

*The authors claim (abstract and conclusion) to make a "spatially resolved estimate ... of glacier volume changes for entire HMA", which would have been very useful since past ICESat studies have been spatially limited or based on older and less accurate versions of the Randolph Glacier Inventory (RGI). However, in the end, there is not a single glacier volume (or mass) change presented here, only figures of spatially averaged elevation trends for regions/zones that do not comply with past publications and RGI, making it impossible for the reader to make out the total numbers. Some aggregated numbers based on upscaling with RGI areas would be highly useful both for comparison with past studies (including GRACE) and as reference for glacier/climate assessments.*

We completely agree with the reviewer. In the revised version, our new zonation, surface elevation changes and corresponding glacier volume changes (using RGI glacier areas) will be available in the supplement.

*Despite these critical points, I do think that this study is highly valuable and should be considered for publication in the Cryosphere after careful revisions. I have listed a number of more specific comments, edits and suggestions in chronological order below. They refer to printed page and line numbers in the discussion manuscript which unfortunately often differ from true page-by-page line numbers.*

*P1, L2: A "diverse pattern" of volume change would be highly dependent on regional glacier area. I think you actually mean elevation changes in this context, so that should be mentioned here or in the previous sentence.*

Changed to "surface elevation changes"

*P1, L3: I find it awkward to say "driven by ... glacier sensitivity". The main physical driver is precipitation changes, but different glaciers can indeed have different sensitivities to that. I suggest to rewrite this sentence.*

Changed to "caused by"

*P1, L6: I think this statement is based on the reanalysis data which are discussed further down in the abstract. It is better to discuss topic-by-topic in a coherent manner.*      DONE

*P1, L13: "Considering evaporation loss, ... ". What do you mean? It sounds like you are not considering it here since you talk about "average annual precipitation". Please clarify.*

Changed to "taking into account"

*P1, L16: Unclear what is meant by "geometry changes". Remove or explain.*      DONE

*P1, L18: Should be past tense, like the rest of the abstract, since it refers to a distinct period (2003-2008). Please check this elsewhere too although it is not a big issue.*      DONE

*P2, L2: Or the "Pamir-Karakoram anomaly" as suggested by Gardelle et al. (2013)*      CHANGED

*P2, L8: reduced/decreased evaporation (for consistency)*      DONE

*P2, L15-19: Some of these studies are not region-wide for HMA, but rather HKKH or Tibet only. That makes this study even more relevant (which could be highlighted).*      DONE

*P2, L29: Any reference(s) for the last two issues?*

Brun et al (2017) themselves mention problems due to too much noise for time periods shorter than a decade as well as the issue of varying studied time periods throughout the area. We added references to all issues.

*P2, L35: You concluded here or Kääb et al. concluded in 2015?*      Kääb et al (2012) – CHANGED

*P3, L3-4: It is not intuitive what "hypsometries of individual years of ICESat samples" and "elevation trend in ... sampling elevations" actually means. I think you should explain what hypsometry is and why it is important in this context, or use different wording to explain what you want to say.*     DONE

*P3, L7: 2018 -> 2008. And either you should explain what was special for this campaign or you should not mention it here.*     DONE

*P3, L15: Write out and reference RGI. And plural - regions of TP and Kunlun Shan?*     DONE

*P3, L20: I don't understand this sentence, and it doesn't seem needed either.*     REMOVED

*P3, L21: "The HMA glacier region is covered by ... "*     DONE

*P4, L2: The fact that extensive parts of HMA has predominant spring/summer accumulation seems to contradict your reasoning to exclude all ICESat winter data ( ~March) because of variable winter snow, at least for some of your zones.*

   This is correct, but we don't see this as a contradiction. In areas with spring/summer accumulation, accumulation and ablation happen at the same time, and some of these regions may still receive a share of their accumulation in winter. Thus, autumn still marks the end of the hydrological year in all areas, and it is the season with least snow cover (and cloud cover) in entire HMA. For a consistent approach we thus only use autumn campaigns. We agree with the reviewer that winter campaigns may be useful in some areas for studies with local character. See also comments for P6, L12 below considering including winter campaign data.

*P5, L9: This is also nicely shown by Kraaijenbrink et al. [2017], but unfortunately hidden in the Supplementary information of the paper.*     ADDED

*P5, L28: Reference for these data?*     ADDED

*P6, L6: 1990s*   CHANGED

*P6, L12: See comment P4-L2. Considering that the ICESat data sampling is very limited, don't you miss out on a lot of potentially good data in TP and southeasterly regions where winters are relatively dry? I agree that the early summer data should be excluded though.*

   As explained above, we think a spatially consistent approach is preferable for this HMA-wide study. Even in very dry areas, we find evidence of winter snow from processing/analysing the data (detailed below). Note that if winter data should be used in other/future studies focusing on e.g. only the driest areas within HMA, autumn and winter trends have to be computed separately to avoid a bias from consistently higher elevations (snow cover) at the end of the studied period. Data points in each end of a trend analysis have a considerably larger influence on trend slopes than other data points.
   Figure 3 shows the modelled snow fall between the first (October) and second (December) half of the autumn 2008 campaign, and it can be seen that only some areas of the TP receive very little precipitation during the first half of the winter. However, the modelled precipitation is highly unsure and underestimates precipitation at higher elevations severely (e.g. by a factor of 4 at Aru glacier in western TP, Kääb et al. 2017, suppl.). We visually analysed optical satellite imagery of the
entire region to classify ICESat laser footprints and for lake areas, and found clear evidence of winter
snow fall on the entire TP. While processing and analysing the data, we always also plotted campaign
medians for winter and early spring campaigns and found in most cases indications of higher surface
elevations during winter also for our zones on the TP, either from snow fall or possibly also ice
emergence at the tongues (due to glacier dynamics).  Analysing the difference between
summer/winter data more closely is thus not feasible within this already extensive project but might
be interesting for a future study.

[Figure]

**Figure 3: MERRA-2 snow precipitation for October, November and December 2008. (We chose snow precipitation rather**
**than total precipitation to better highlight the mountain areas – otherwise rain fall in the lowlands dominate the plot. At**
**the elevations of question, temperatures during the cold season are low enough that all precipitation falls as snow).**

*P6, L8: glacier samples*

We are not sure we understand what the reviewer means here, but guess (s)he wants to
make sure readers not familiar with ICESat glacier analyses understand what data we use. We thus
rewrote the sentence.

*P6, L17: Any name(s)/reference(s) for the clustering methods you tested?*

Unfortunately not – we researched common clustering methods and realised quickly that
applying these to our problem would require building a new and very complex iterative setup that 1)
groups our data, then 2) runs all kind of analyses to check how similar the data point attributes are
not only in spatial space, but in terms of topographic parameters (elevation, aspect…), glacier
attributes (glacier type, collecting all points from one glacier within one group) and setting (climate,
orography…) and possibly also the resulting elevation trend and uncertainty. Setting up this model
would have been extremely challenging as many of the attributes above are not hard facts and would
require some kind of categorising process themselves, and the iterative approach would be much too
costly computationally. We thus got stuck very early in implementing (inventing) such an automated
clustering, and rather did the same thing manually. Which spatial clustering algorithm we would have
chosen in the above setup would have been rather irrelevant compared to a lot of other choices and
definitions needed, so we prefer not to refer to any methods.

However, we still think it is useful to mention that we thought of this possibility and had to
abandon it, as other researchers might have the same idea. We changed the wording from "tested"
to "considered" and better explain how the zonation was made in the corresponding appendix.

*P6, L21. I think this is a good way to do it.*

We appreciate the reviewer's approval.

*P6, L26: Reference or explanation for the four methods?*

We moved the reference to the appendix further up in the text. The methods are described
there.

*P6, L32: This paragraph is confusing and the correction needs to be better warranted. What is*
*actually meant by "local reference elevation bias"? If a bias is truly local, then it is rather a local*
*error (not systematic). In this case, would it not be better termed a "glacier-by-glacier bias"? But*
*then comes the question of what causes such biases and why a correction is needed. Different DEM*
*source date between glaciers within the same region is an obvious explanation in Treichler and Kääb*
*(2016), but that is less of an issue here since the SRTM DEM stems from a single year.*
*Instead, a glacier-median dh correction might erroneously mute some of the real elevation change*
*signal because the glacier-median time of ICESat samples will also vary from glacier to glacier.*

See answer to the next comment

*P6, L33: Why just from "snow fall in the second part of the autumn 2008 campaign"? If I understand*
*your correction right, the "glacier-by-glacier bias" is impacted by any type of elevation change*
*between the time of SRTM and the respective ICESat measurements? I see the variation in glacier-*
*median dh as a result of variable temporal and altitudinal sampling of ICESat between glaciers, as*
*well as various errors in the SRTM DEM. If the latter is the main issue, why not use nearby land-*
*samples to determine this correction?*

As the reviewer writes below, most of the confusion seems to have been solved by the text
in the appendix. With local reference bias, we mean any elevation error in the reference DEM that is
not systematic (i.e. cannot be removed in a systematic way such as processing per DEM tile). In case
of the SRTM DEM this may be horizontal/vertical shifts of single InSAR scenes of penetration into
snow, ice or sand. The per-glacier correction was introduced in Treichler and Kääb (2016) to correct
for sub-tile misregistration and DEM age. In the case of the SRTM, local shifts/misregistration effects
were clearly visible, too – but on glaciers, we found penetration to be the dominating effect; in
particular in dry accumulation areas e.g. on the TP. This is also visible in the very steep dh-elevation
gradients in this area (fig. 3 d). The reviewer is correct in that such a correction may remove some of
the elevation change signal. In the study of Treichler and Kääb (2016), the benefits of the correction
outweighed the "flattened" trend by far, as only corrected ICESat data followed the annual
cumulative mass balance curve of southern Norway's glaciers. This is also the case for many of our
HMA spatial units, but the large scope of the study doesn't allow for discussion of each single spatial
unit to justify individually applied corrections. We thus only apply the correction where it made the
trends steeper (see appendix), i.e. the correction clearly outweighed its side effects of "flattening"
the elevation change signal.
The correction for December 2008 snow fall is independent of the per-glacier correction. It is
computed using nearby land samples, as the author suggests. We use the elevation difference (i.e.

snow layer) between the 2008 October and December campaigns and compute a snow depth/elevation relationship, assuming a linear increase of snow depth with elevation.

We rewrote the paragraph to make it clearer. In particular, we point to the appendix earlier in the text, and better separate the two different corrections (per-glacier correction and December 2008 snow fall).

*P6, L35: What is the correction applied to? I understand it as a normalization of dh on a glacier-by-glacier basis by subtracting the median dh for each glacier, but this is not clear.*

This is exactly how the correction works. We hope that the changes made to the paragraph (reply above) and the explanations in the appendix explain this well enough.

*P7, L13: Do you actually mean non-glacier mass changes? Hence, removing the gravity signal from changing lakes to derive glacier mass changes from GRACE. Please clarify.*

Yes. Resolving the gravimetry signal for a specific component (e.g. glacier mass changes) requires that all other contributing factors are known (changes in surface water, ground water, permafrost, biomass…). We rewrote the sentence so that it becomes clearer that we don't include gravimetry in this study but rather see this as a potential application.

*P7, L24: references?*     ADDED

*P8, L5-7: Long and complicated sentence.*
*P8, L5-10:  This section doesn't really describe a clear method beyond looking at the data and taking decadal averages. Is it needed? More confusing than clarifying.*

True. We removed the paragraph.

*P8, L13: 100 units? It doesn't look like so many.*

The number is correct, there are 100 units.

*P8, l17: What is done with those 34 units and why?*

For these units, we applied the cG correction to dh and/or used only hypsometry methods C and D due to systematic hypsometry missampling (consistently too high/low elevations sampled). The methods and appendices have been rewritten to clarify which corrections were applied where, and why.

*P8, L21: This correction appears out of nowhere. Remove or reference appendix B3.*

We hope that the correction now is better explained in the method section. We added a reference to the appendix.

*P8, L24: Delete last sentence (already explained)*     DONE

*P8, L28: Interesting point, but since the grid cells are already overlapping by 50% and will be naturally smoothed by that, the conclusion is weak.*

That is in principle true, but the pattern is smoothed even if grid cells are not overlapping – this is visible Brun et al. 2017 (1x1 degree grid cells there, and not overlapping due to spatially denser results with ASTER DEM stacks). We thus think our conclusion is correct and left it as is, but added a
reference to Brun et al. 2017.

*P10, L1: Nice!*

*P10, L35: I do not fully agree. See comments P6-L33 and P30-L20.*

See answers to the above comments.

*P12, L7: Any idea why?*

This is the case for all types of linear fits, and the reason why we used three different fitting
methods that are supposedly least sensitive to "outliers" or extreme values in either end of the time
axis. We don't have a statistical proof at hand, but rather try to explain this with an example: Given a
linear trend with samples that fit the line relatively well. Take one (or even a whole bunch of samples)
a) from the middle of the time axis or b) from the very beginning or end of the time axis, and offset it
substantially. Fitting a linear trend through a) will not change the trend but only increase the trend
error. For b), however, the trend line slope will be increased/decreased to better incorporate the
sample(s). That's how the statistical models work, and what they were designed for. Thus, bias from
snow fall in the very end of the timeline affects the trends much more than any bias in the middle of
the time axis (see also Appendix D in the paper). With this example, we want to illustrate how
unexpectedly big such an effect can be – something scientist maybe should be more aware of since
linear fits are commonly used (and little questioned) for time series, in particular in relation to
climate change.

*Fig. 3: Interesting figure, but I suggest to use other colors for panel c to avoid confusion with the*
*thickening-thinning colors in a-b.*

We will adapt the figure accordingly for resubmission.

*P12, L10: Also mentioned in the caption. Once is enough.*          REMOVED

*P12, L15. I don't think this specification is needed.*

We prefer to keep this specification, as the y-axis measure is somewhat non-intuitive. We are
afraid readers might easily mistake the graph and assume it shows volume changes (i.e. lake volumes
are doubled, halved…), but our graph rather shows what share of the (total) volume change
happened when. We shortened and reworded the sentence to make this clearer.

*Fig. 4b: Label regions according to Table 1.*

We will adapt the figure accordingly for resubmission.

*Table 1:  The caption is rather confusing, listing three time spans next to each other (belonging  to*
*different  columns  of  the  table)  and  giving  volume  change  in  unit  mm  without  describing  if  it  is*
*per lake area or basin area.*

We rewrote the caption to clarify timespans and volume change units (per basin area).

*Fig 6:  The combination of two stations in the upper panel makes this figure unnecessarily hard to*
*read. I suggest to split them in each their panel.*

We will adapt the figure accordingly for resubmission.

*Fig. 7: Why does panel-a show ERA-Interim summer and panel-b MERRA-2 annual, and not either*
*the same period for both products or both periods for one product. Also, the figure is only discussed*
*very briefly, and well after Fig. 8 in the text. I think the figure is interesting, but to be included, it*
*should be properly referenced and discussed in the text.*

The reviewer is right – we accidentally added the wrong figure in the manuscript. Fig 7b has
now been replaced with difference in summer P, i.e. same periods as for ERA Interim. The spatial
pattern of the replaced figure is however nearly the same, in particular for the regions with increased
precipitation – which indicates that the precipitation increase happened during summer months. We
changed the order of the figures and refer to them earlier and more thoroughly in the text.

*Fig. 8: The P-E curves appear are faded and hard to see, despite being most relevant in theory. I*
*suggest you use a thicker line or sharper color/tone to improve visibility.*

We will adapt the figure accordingly for resubmission.

*P16, L6. Specify these regions (NE, NW, C)*    DONE

*P16, L8: considering the high uncertainty; "results in" -> "suggests"*
*P16, L9: Fig. 8b?*

The paragraph has been rewritten.

*P19, L20: ... between the periods*    DONE

*P20, L23: "over 1988-2007" or "between 1988 and 2007"*    DONE

*P21, L18: This sentence is difficult to understand.*    REWRITTEN

*P21, L16-29: I would expect this paragraph to be closer linked with the interesting Fig. 3, as well as*
*independent studies of velocity changes, of which the recently published study of Dehecq et al. [2019]*
*seems particularly relevant.*

We agree with the reviewer that the interesting dynamical aspects rather got too little room
in our discussion. We thus rewrote the paragraph to discuss Fig. 3 and its implications in more detail,
and related our findings to the velocity changes published by Dehecq et al. (2019) while this paper
was in review. Unfortunately, the time period (2000-2016) and spatial aggregation doesn't allow a
very detailed comparison with our results.

*P22, L13: Move authors out of the parentheses.*    DONE

*P22, L22: I think you really want to talk about elevation changes (or thinning) here since actual*
*volume changes are so dependent on regional glacier area.*    REPHRASED

*P22, L31. Unclear sentence.*    REWRITTEN

*P22, L12: use abbreviated m w.e. a-1, as elsewhere*    DONE

*P22, L22: Is "glacier sensitivity to precipitation" an appropriate heading for this section? I feel it is more a discussion of orographic effects that cause different precipitation regimes on either side of a ridge, not really whether glaciers are more or less sensitive to precipitation in general. Or you need to better explain what you mean by "sensitivity".*

We changed the section heading to "Glacier mass balance and precipitation…".

*P23, L30: ... both surging glaciers and glaciers recovering ...*     DONE

*P23, L32: Combine references with same authors.*        DONE

*P23, L8: This paragraph is very detailed compared to the others and could be shortened.*

The paragraph has been shortened.

*P25, L20: The Conclusions section provides a good summary, but would benefit from a shortening to better highlight the main findings and outlook.*

We rewrote the conclusions to better highlight the main findings.

*P25, L21: This study has many new and interesting aspects, but it is not the first one and does not actually present any volume changes. I would rather highlight the improved zoning and joint analysis of lake changes as the most unique part of this paper.*

The reviewer is right in that our study is not the first one, but the region indeed lacked a complete, consistent analysis of ICESat data so far – all other studies only looked at parts of the region and/or used methods that are affected by biases. We rewrote the paragraph to rather highlight the consistent approach and the aspects the reviewer proposes.

*P25, L11: The selective mention of MERRA-2 (which fits with your observations) and not ERA-Interim (which doesn't fit) is peculiar as long as you cannot identify reasons why one product should be better than the other. Mention both products or none.*

We agree with the reviewer and added a statement about ERA Interim.

*P26, L15: Is this significant? If not, no need to mention as a conclusion.*

Many studies that assess lake changes claim that this is due to increased lake influx from glacier melt. Our study shows that this is likely a very small contributing factor. We thus think it is important to mention this in the conclusions.

*P26, L10: Since Cryosphere has easy support for auxiliary data, it would be very nice for the community if the zoning (including glacier area and averaged dh/dt) is provided with the paper, not only on personal request.*

We'll include the zoning and glacier surface change rates as auxiliary data supplement.

*P28, L18: ICESat period*        DONE

*P30, L18: Method A is not clear.*

We added more explanations for method A and a reference to Kääb et al. (2012), where the method was first described.

*P30, L20: Doesn't the B correction also introduce an error due to ICESat's variable temporal sampling? I.e., if a glacier is thinning, then ICESat observations in the later years of the mission would naturally have lower dh values than expected from the general dh-elevation trend. This is the same issue as pointed out for the cG correction (see comment P6-L32). Are both corrections applied in the case of method B?*

This is a good observation but not true, as the two methods are different in that aspect. In contrary to the cG correction, method B uses a dh—elevation gradient that is computed from all samples within a spatial unit and applied to all samples, no matter which glacier or campaign. The gradient will be the same for all campaigns, and be in the order of 0-3 m per 100 m elevation (fig. 3d in the paper). Assuming glacier thinning (predominantly on the tongues), the real slope of the dh— elevation gradient would be less (more) in the beginning (end) of the studied time period, resulting in residuals – which correspond to the signal: We use that gradient to remove the expected elevation dependency so that only the temporal aspect remains. In other words, after correction using method B, the regression is done on dh anomalies – which are the (local) thinning/thickening signal (plus other local biases). For method B, the dh—elevation gradient thus has to be the same for the entire time period and the method does thus not introduce an error, but instead removes noise to better show the thinning signal in the example e reviewer states.
Elevation correction (methods A-D) is done everywhere. cG is applied only in some cases (see answers to the other reviewer comments concerning vertical bias correction).

*P30, L21: What is meant by "filtering"? If you mean removing/culling data, then useful observations would also be removed and I see no reason for that as long as you can rather introduce a weighting scheme like Method D.*

Yes, we mean removing of data. All methods (A-D) have the same goal, to remove elevation-induced bias and false trends due to changing sampling elevations with time. It might seem a bad idea to remove data, but statistically seen, the weighting does essentially the same as it removes equally much influence of "good" samples. Depending on local sampling (timing, location, elevations, also for/within each individual glacier…) in each spatial unit, some of the methods are more appropriate than others, and some would even fail (i.e. introduce errors) for some spatial units. For the vast majority, the differences in final trend estimates lie within 0.1 m/a between methods A-D, if only one method is applied. Using the average of all four methods is thus an appropriate choice to ensure consistency for our study while minimising errors potentially introduced by one of the methods. For more local applications (one spatial unit), however, we recommend to assess all methods carefully.
In fact, we analysed all four methods and how they differ thoroughly for all spatial units – but these technical details and comparisons don't fit this paper (and the journal) well, have negligible influence on the results and are of little importance for the focus of the study, so we prefer to not add more details here. The interested reader will find an extended discussion of all methods, corrections and their implications on the results in the PhD thesis of Treichler (2017).

*P30, L10: Thanks, this shows that you are aware of my issue with the cG correction and method B. Since it is in the end only applied to 6 units – is it really needed? And what about the same issue for Method B?*

The cG correction was first proposed by Treichler & Kääb (2016), where also its potential side effect of flattening the elevation change trend slope is discussed. In that study, the benefits clearly outweighed the side effect, as campaign medians followed southern Norway's cumulative mass balance evolution only after correction. For many spatial units in this study, the same is the case. However, the correction might not seem equally important since we don't assess campaign medians but only trend slopes. Nevertheless, the fact that some of the elevation change rates become steeper after cG correction (while the contrary is expected) makes it clear that the correction should be applied for these units. Besides, we would like to keep the correction also in this study to encourage its use in other studies, in particular studies of more local character. However, the number 6 stated in the appendix was wrong – assumingly, this was a crippled 16, as we applied cG to 16 units, only hypsometry correction methods C and D to 13 units, and both of the above to 5 units. This was not sufficiently explained in the manuscript, and it also seems errors were not for all 34 cases completely propagated (i.e. the total error should include trend slope differences for with/without applied correction). We double-checked for consistency and updated the corresponding trend errors.

In the revised paper, the methods and appendices are rewritten.

*P32, L24: Standard error of the mean?* yes – CHANGED

[revised manuscript text omitted]

---

## Author Response (AR2)

**Response to referee comments**

**Recent glacier and lake changes in High Mountain Asia and their relation to precipitation changes**

Désirée Treichler, Andreas Kääb, Nadine Salzmann, and Chong-Yu Xu

The Cryosphere Discussion, doi: 10.5194/tc-2018-238

We would like to thank the reviewer and editor for their thorough work that greatly helped to improve the article. All suggestions have been implemented as proposed – except for two. These are outlined below, together with a mark-up manuscript version where the changes made in response to the referees' comments are highlighted.

**Anonymous Referee #2**

*Dear authors,*

*thank you for a very thorough revision which I think has made the paper much more readable and clear. It was nice to read it again, and I am happy to recommend it for publication. I only have some smaller technical corrections given below, and a suggestion to carefully consider what wording is most correct when you talk about changes in lake volumes and lake levels. In many cases, I think the latter would be more appropriate since you refer to area-specific numbers as for precipitation and glacier mass balance.*

*Technical corrections and comments to the revised manuscript version:*

*P2, L20: Remove sentence or make it correct; Garner et al. (2013) and Brun et al. 2017*     DONE

*P3, L14-15: 2018 -> 2008*     DONE

*P10, L3: volume changes -> mass changes (since unit is Gt a-1). Please check consistency elsewhere also.*     DONE

*Section 3.2: The title says "...volume change" although the text only concerns elevation trends. Please change the header accordingly or add brief text on how you derived volume change (or better; mass balance/change as in tables/supplement).*     DONE

*Section 4.2: This is a discussion of errors rather than glaciology/climate as in the other subsections, so I think that should also be reflected in the section title, e.g. "4.2 Biasing influences from..." (consistent with Appendix D)*     DONE

*P13, caption: Are circles scaled by area or radius/diameter. Please specify in caption.* DONE

*Table 1, caption: It seems strange to use volume and dV as temrs when numbers are provided as area*
*specific rates? I rather think of it as basin-averaged lake level changes. If changed, please make sure*
*that terms become consistent throughout the manuscript.*

**From the reviewer's last comment, we are unsure whether we explained the units in the table (and**
**possibly figure 4) well enough - the table does not show average lake level changes but changes in**
**lake water volumes converted to mm m$^{-1}$, i.e. spread across the entire catchment area. "Area-**
**specific" volumes/rates might be a difficult term as it is not necessarily obvious for all readers**
**which area we refer to (lake or catchment). The conversion we did is introduced in the method**
**section 3.3 and appendix C2. The reason for providing catchment-area-specific numbers is to show**
**differences between the regions - which can be achieved by using the same units as for**
**precipitation (mm m$^{-2}$). Plain lake level changes are highly depending on individual lake and**
**catchment sizes. Thus, we think "basin-averaged lake level changes" could also be rather**
**misleading. Instead, we adapted some of the text in the table and figure caption and**
**corresponding paragraph to better explain and distinguish the numbers shown here and in the**
**previous figures.**

**Comments from the Editor**

*Corrections were made directly in the track changes document.*

**ALL DONE**
**except:**

*P2, L22 (track changes document): Zemp, 2019?*

*P2, L24: self promotion, but you could also include Wouters et al, Frontiers, 2019 here, which is an*
*update of the GRACE time series in Gardner, 2019*

**We acknowledge the scientific value of these recent studies but nevertheless prefer not to include**
**them. The reason is that these are global-scale rather than regional-scale studies, without specific**
**focus on our study region. The currently cited studies were chosen as they provide a region-wide**
**assessment of the area and also explicitly show/discuss spatial differences in HMA glacier**
**evolution, which is not usually the case for global-scale studies.**

[revised manuscript text omitted]